# Uniform Convergence of Gradients for Non-Convex Learning and Optimization

**Dylan J. Foster**
Cornell University
djfoster@cornell.edu

**Ayush Sekhari**
Cornell University
sekhari@cs.cornell.edu

**Karthik Sridharan**
Cornell University
sridharan@cs.cornell.edu

## Abstract

We investigate 1) the rate at which refined properties of the empirical risk—in particular, gradients—converge to their population counterparts in standard non-convex learning tasks, and 2) the consequences of this convergence for optimization. Our analysis follows the tradition of *norm-based capacity control*. We propose vector-valued Rademacher complexities as a simple, composable, and user-friendly tool to derive *dimension-free* uniform convergence bounds for gradients in non-convex learning problems. As an application of our techniques, we give a new analysis of batch gradient descent methods for non-convex generalized linear models and non-convex robust regression, showing how to use any algorithm that finds approximate stationary points to obtain optimal sample complexity, even when dimension is high or possibly infinite and multiple passes over the dataset are allowed.

Moving to non-smooth models we show—-in contrast to the smooth case—that even for a single ReLU it is not possible to obtain dimension-independent convergence rates for gradients in the worst case. On the positive side, it is still possible to obtain dimension-independent rates under a new type of distributional assumption.

## 1 Introduction

The last decade has seen a string of empirical successes for gradient-based algorithms solving large scale non-convex machine learning problems [24, 16]. Inspired by these successes, the theory community has begun to make progress on understanding when gradient-based methods succeed for non-convex learning in certain settings of interest [17]. The goal of the present work is to introduce learning-theoretic tools to—in a general sense—improve understanding of when and why gradient-based methods succeed for non-convex learning problems.

In a standard formulation of the non-convex statistical learning problem, we aim to solve

$$\text{minimize} \quad L_{\mathcal{D}}(w) \coloneqq \mathbb{E}_{(x,y)\sim\mathcal{D}} \, \ell(w \, ; x, y),$$

where $w \in \mathcal{W} \subseteq \mathbb{R}^d$ is a parameter vector, $\mathcal{D}$ is an unknown probability distribution over the instance space $\mathcal{X} \times \mathcal{Y}$, and the loss $\ell$ is a potentially non-convex function of $w$. The learner cannot observe $\mathcal{D}$ directly and instead must find a model $\widehat{w} \in \mathcal{W}$ that minimizes $L_{\mathcal{D}}$ given only access to i.i.d. samples $(x_1, y_1), \ldots, (x_n, y_n) \sim \mathcal{D}$. Their performance is quantified by the *excess risk* $L_{\mathcal{D}}(\widehat{w}) - L^\star$, where $L^\star = \inf_{w\in\mathcal{W}} L_{\mathcal{D}}(w)$.

Given only access to samples, a standard ("sample average approximation") approach is to attempt to minimize the *empirical risk* $\widehat{L}_n(w) \coloneqq \frac{1}{n} \sum_{t=1}^n \ell(w \, ; x_t, y_t)$. If one succeeds at minimizing $\widehat{L}_n$, classical statistical learning theory provides a comprehensive set of tools to bounds the excess risk of the procedure. The caveat is that when $\ell$ is non-convex, global optimization of the empirical risk may be far from easy. It is not typically viable to even *verify* whether one is at a global minimizer

of $\widehat{L}_n$. Moreover, even if the population risk $L_{\mathcal{D}}$ has favorable properties that make it amenable to gradient-based optimization, the empirical risk may not inherit these properties due to stochasticity. In the worst case, minimizing $L_{\mathcal{D}}$ or $\widehat{L}_n$ is simply intractable. However, recent years have seen a number of successes showing that for non-convex problems arising in machine learning, iterative optimizers can succeed both in theory and in practice (see [17] for a survey). Notably, while minimizing $\widehat{L}_n$ might be challenging, there is an abundance of gradient methods that provably find approximate stationary points of the empirical risk, i.e. $\|\nabla \widehat{L}_n(w)\| \le \varepsilon$ [33, 10, 38, 3, 26]. In view of this, the present work has two aims: *First*, to provide a general set of tools to prove uniform convergence results for gradients, with the goal of bounding how many samples are required to ensure that with high probability over samples, simultaneously over all $w \in \mathcal{W}$, $\|\nabla L_{\mathcal{D}}(w)\| \le \|\nabla \widehat{L}_n(w)\| + \varepsilon$; *Second*, to explore concrete non-convex problems where one can establish that the excess risk $L_{\mathcal{D}}(\widehat{w}) - L^\star$ is small a consequence of this gradient uniform convergence. Together, these two directions yield direct bounds on the convergence of non-convex gradient-based learning algorithms to low excess risk.

Our precise technical contributions are as follows:

- We bring vector-valued Rademacher complexities [30] and associated vector-valued contraction principles to bear on the analysis of uniform convergence for gradients. This approach enables *norm-based capacity control*, meaning that the bounds are independent of dimension whenever the predictor norm and data norm are appropriately controlled. We introduce a "chain rule" for Rademacher complexity, which enables one to decompose the complexity of gradients of compositions into complexities of their components, and makes deriving dimension-independent complexity bounds for common non-convex classes quite simple.

- We establish variants of the *Gradient Domination* condition for the population risk in certain non-convex learning settings. The condition bounds excess risk in terms of the magnitude of gradients, and is satisfied in non-convex learning problems including generalized linear models and robust regression. As a consequence of the gradient uniform convergence bounds, we show how to use any algorithm that finds approximate stationary points for smooth functions in a black-box fashion to obtain optimal sample complexity for these models—both in high- and low-dimensional regimes. In particular, standard algorithms including gradient descent [33], SGD [10], Non-convex SVRG [38, 3], and SCSG [26] enjoy optimal sample complexity, even when allowed to take multiple passes over the dataset.

- We show that for non-smooth losses dimension-independent uniform convergence is not possible in the worst case, but that this can be circumvented using a new type of margin assumption.

**Related Work**    This work is inspired by [31], who gave dimension-dependent gradient and Hessian convergence rates and optimization guarantees for the generalized linear model and robust regression setups we study. We move beyond the dimension-dependent setting by providing norm-based capacity control. Our bounds are independent of dimension whenever the predictor norm and data norm are sufficiently controlled (they work in infinite dimension in the $\ell_2$ case), but even when the norms are large we recover the optimal dimension-dependent rates.

Optimizing the empirical risk under assumptions on the population risk has begun to attract significant attention (e.g. [12, 18]). Without attempting a complete survey, we remark that these results typically depend on dimension, e.g. [18] require $\mathrm{poly}(d)$ samples before their optimization guarantees take effect. We view these works as complementary to our norm-based analysis.

**Notation**    For a given norm $\|\cdot\|$, the dual norm is denoted $\|\cdot\|_\star$. $\|\cdot\|_p$ represents the standard $\ell_p$ norm on $\mathbb{R}^d$ and $\|\cdot\|_\sigma$ denotes the spectral norm. $\mathbf{1}$ denotes the all-ones vector, with dimension made clear from context. For a function $f : \mathbb{R}^d \to \mathbb{R}$, $\nabla f(x) \in \mathbb{R}^d$ and $\nabla^2 f(x) \in \mathbb{R}^{d \times d}$ will denote the gradient and the Hessian of $f$ at $x$ respectively. $f$ is said to be $L$-Lipschitz with respect to a norm $\|\cdot\|$ if $|f(x) - f(y)| \le L\|x - y\| \ \forall x, y$. Similarly, $f$ is said to be H-smooth w.r.t norm $\|\cdot\|$ if its gradients are H-Lipschitz with respect to $\|\cdot\|$, i.e. $\|\nabla f(x) - \nabla f(y)\|_\star \le H\|x - y\|$ for some $H$.

## 2 Gradient Uniform Convergence: Why and How

### 2.1 Utility of Gradient Convergence: The Why

Before introducing our tools for establishing gradient uniform convergence, let us introduce a family of losses for which this convergence has immediate consequences for the design of non-convex statistical learning algorithms.

**Definition 1.** *The population risk $L_{\mathcal{D}}$ satisfies the $(\alpha, \mu)$-Gradient Domination condition with respect to a norm $\|\cdot\|$ if there are constants $\mu > 0$, $\alpha \in [1, 2]$ such that*

$$L_{\mathcal{D}}(w) - L_{\mathcal{D}}(w^\star) \le \mu \|\nabla L_{\mathcal{D}}(w)\|^\alpha \quad \forall w \in \mathcal{W}, \tag{GD}$$

*where $w^\star \in \arg\min_{w \in \mathcal{W}} L_{\mathcal{D}}(w)$ is any population risk minimizer.*

The case $\alpha = 2$ is often referred to as the Polyak-Łojasiewicz inequality [36, 23]. The general GD condition implies that all critical points are global, and is itself implied (under technical restrictions) by many other well-known conditions including one-point convexity [28], star convexity and $\tau$-star convexity [14], and so-called "regularity conditions" [44]; for more see [23]. The GD condition is satsified—sometimes locally rather than globally, and usually under distributional assumptions— by the population risk in settings including neural networks with one hidden layer [28], ResNets with linear activations [13], phase retrieval [44], matrix factorization [29], blind deconvolution [27], and—as we show here—generalized linear models and robust regression.

The GD condition states that to optimize the population risk it suffices to find a (population) stationary point. What are the consequences of the statement for the learning problem, given that the learner only has access to the empirical risk $\widehat{L}_n$ which itself may not satisfy GD? The next proposition shows, via gradient uniform convergence, that GD is immediately useful for non-convex learning even when it is only satisfied at the population level.

**Proposition 1.** Suppose that $L_{\mathcal{D}}$ satisfies the $(\alpha, \mu)$-GD condition. Then, for any $\delta > 0$, with probability at least $1 - \delta$ over the draw of the data $\{(x_t, y_t)\}_{t=1}^n$, every algorithm $\widehat{w}^{\mathrm{alg}}$ satisfies

$$L_{\mathcal{D}}(\widehat{w}^{\mathrm{alg}}) - L^\star \le 2\,\mu\left(\left\|\nabla\widehat{L}_n(\widehat{w}^{\mathrm{alg}})\right\|^\alpha + \mathbb{E}\sup_{w \in \mathcal{W}}\left\|\nabla\widehat{L}_n(w) - \nabla L_{\mathcal{D}}(w)\right\|^\alpha + c\left(\frac{\log\left(\frac{1}{\delta}\right)}{n}\right)^{\frac{\alpha}{2}}\right), \tag{1}$$

where the constant $c$ depends only on the range of $\|\nabla\ell\|$.

Note that if $\mathcal{W}$ is a finite set, then standard concentration arguments for norms along with the union bound imply that $\mathbb{E}\sup_{w \in \mathcal{W}}\|\nabla\widehat{L}_n(w) - \nabla L_{\mathcal{D}}(w)\| \le O\left(\sqrt{\frac{\log|\mathcal{W}|}{n}}\right)$. For smooth losses, if $\mathcal{W} \subset \mathbb{R}^d$ is contained in a bounded ball, then by simply discretizing the set $\mathcal{W}$ up to precision $\varepsilon$ (with $O(\varepsilon^{-d})$ elements), one can easily obtain a bound of $\mathbb{E}\sup_{w \in \mathcal{W}}\|\nabla\widehat{L}_n(w) - \nabla L_{\mathcal{D}}(w)\| \le O\left(\sqrt{\frac{d}{n}}\right)$. This approach recovers the dimension-dependent gradient convergence rates obtained in [31].

Our goal is to go beyond this type of analysis and provide dimension-free rates that apply even when the dimension is larger than the number of examples, or possibly infinite. Our bounds take the following "norm-based capacity control" form: $\mathbb{E}\sup_{w \in \mathcal{W}}\|\nabla\widehat{L}_n(w) - \nabla L_{\mathcal{D}}(w)\| \le O\left(\sqrt{\frac{\mathcal{C}(\mathcal{W})}{n}}\right)$ where $\mathcal{C}(\mathcal{W})$ is a norm-dependent, but dimension-independent measure of the size of $\mathcal{W}$. Given such a bound, any algorithm that guarantees $\|\nabla\widehat{L}_n(\widehat{w}^{\mathrm{alg}})\| \le O\left(\frac{1}{\sqrt{n}}\right)$ for a $(\alpha, \mu)$-GD loss will obtain an overall excess risk bound of order $O\left(\frac{\mu}{n^{\alpha/2}}\right)$. For $(1, \mu_1)$-GD this translates to an overall $O\left(\frac{\mu_1}{\sqrt{n}}\right)$ rate, whereas $(2, \mu_2)$-GD implies a $O\left(\frac{\mu_2}{n}\right)$ rate. The first rate becomes favorable when $\mu_1 \le \sqrt{n} \cdot \mu_2$ which typically happens for very high dimensional problems. For the examples we study, $\mu_1$ is related only to the radius of the set $\mathcal{W}$, while $\mu_2$ depends inversely on the smallest eigenvalue of the population covariance and so is well-behaved only for low-dimensional problems unless one makes strong distributional assumptions.

An important feature of our analysis is that we need to establish the GD condition only for the population risk; for the examples we consider this is easy as long as we assume the model is well-specified. Once this is done, our convergence results hold for *any* algorithm that works on the dataset

$\{(x_t, y_t)\}_{t=1}^n$ and finds an approximate first-order stationary point with $\|\nabla \widehat{L}_n(\widehat{w}^{\mathrm{alg}})\| \leq \varepsilon$. First-order algorithms that find approximate stationary points assuming only smoothness of the loss have enjoyed a surge of recent interest [10, 38, 3, 1], so this is an appealing proposition.

## 2.2 Vector Rademacher Complexities: The How

The starting point for our uniform convergence bounds for gradients is to apply the standard tool of symmetrization—a vector-valued version, to be precise. To this end let us introduce a *normed* variant of Rademacher complexity.

**Definition 2** (Normed Rademacher Complexity). *Given a vector valued class of function $\mathcal{F}$ that maps the space $\mathcal{Z}$ to a vector space equipped with norm $\|\cdot\|$, we define the normed Rademacher complexity for $\mathcal{F}$ on instances $z_{1:n}$ via*

$$\mathfrak{R}_{\|\cdot\|}(\mathcal{F}; z_{1:n}) := \mathbb{E}_\epsilon \sup_{f \in \mathcal{F}} \left\| \sum_{t=1}^n \epsilon_t f(z_t) \right\|. \tag{2}$$

With this definition we are ready to provide a straightforward generalization of the standard real-valued symmetrization lemma.

**Proposition 2.** *For any $\delta > 0$, with probability at least $1 - \delta$ over the data $\{(x_t, y_t)\}_{t=1}^n$,*

$$\mathbb{E} \sup_{w \in \mathcal{W}} \left\| \nabla \widehat{L}_n(w) - \nabla L_{\mathcal{D}}(w) \right\| \leq \frac{4}{n} \mathfrak{R}_{\|\cdot\|}(\nabla \ell \circ \mathcal{W}; x_{1:n}, y_{1:n}) + c \left( \frac{\log\left(\frac{1}{\delta}\right)}{n} \right), \tag{3}$$

*where the constant $c$ depends only on the range of $\|\nabla \ell\|$.*

To bound the complexity of the gradient class $\nabla \ell \circ \mathcal{W}$, we introduce a chain rule for the normed Rademacher complexity that allows to easily control gradients of composition of functions.

**Theorem 1** (Chain Rule for Rademacher Complexity). *Let sequences of functions $G_t : \mathbb{R}^K \to \mathbb{R}$ and $F_t : \mathbb{R}^d \to \mathbb{R}^K$ be given. Suppose there are constants $L_G$ and $L_F$ such that for all $1 \leq t \leq n$, $\|\nabla G_t\|_2 \leq L_G$ and $\sqrt{\sum_{k=1}^K \|\nabla F_{t,k}(w)\|^2} \leq L_F$. Then,*

$$\frac{1}{2} \mathbb{E}_\epsilon \sup_{w \in \mathcal{W}} \left\| \sum_{t=1}^n \epsilon_t \nabla(G_t(F_t(w))) \right\| \leq L_F \mathbb{E}_\epsilon \sup_{w \in \mathcal{W}} \sum_{t=1}^n \langle \boldsymbol{\epsilon}_t, \nabla G_t(F_t(w)) \rangle + L_G \mathbb{E}_\epsilon \sup_{w \in \mathcal{W}} \left\| \sum_{t=1}^n \nabla F_t(w) \boldsymbol{\epsilon}_t \right\|, \tag{4}$$

*where $\nabla F_t$ denotes the Jacobian of $F_t$, which lives in $\mathbb{R}^{d \times K}$, and $\boldsymbol{\epsilon} \in \{\pm 1\}^{K \times n}$ is a matrix of Rademacher random variables with $\boldsymbol{\epsilon}_t$ denoting the $t$th column*

The concrete learning settings we study—generalized linear models and robust regression—all involve composing non-convex losses and non-linearities or transfer functions with a linear predictor. That is, $\ell(w; x_t, y_t)$ can be written as $\ell(w; x_t, y_t) = G_t(F_t(w))$ where $G_t(a)$ is some $L$-Lipschitz function that possibly depends on $x_t$ and $y_t$ and $F_t(w) = \langle w, x_t \rangle$. In this case, the chain rule for derivatives gives us that $\nabla \ell(w; x_t, y_t) = G'_t(F_t(w)) \cdot \nabla F_t(w) = G'_t(\langle w, x_t \rangle) x_t$. Using the chain rule (with $K = 1$), we conclude that

$$\mathbb{E}_\epsilon \sup_{w \in \mathcal{W}} \left\| \sum_{t=1}^n \epsilon_t \nabla \ell(w; x_t, y_t) \right\| \leq \mathbb{E}_\epsilon \left[ \sup_{w \in \mathcal{W}} \sum_{t=1}^n \epsilon_t G'_t(\langle w, x_t \rangle) \right] + L \cdot \mathbb{E}_\epsilon \left\| \sum_{t=1}^n \epsilon_t x_t \right\|.$$

Thus, we have reduced the problem to controlling the Rademacher average for a real valued function class of linear predictors and controlling the vector-valued random average $\mathbb{E}_\epsilon \|\sum_{t=1}^n \epsilon_t x_t\|$. The first term is handled using classical Rademacher complexity tools. As for the second term, it is a standard result ([35]; see [19] for discussion in the context of learning theory) that for all smooth Banach spaces, and more generally Banach spaces of Rademacher type 2, one has $\mathbb{E}_\epsilon \|\sum_{t=1}^n \epsilon_t x_t\| = O(\sqrt{n})$; see Appendix A for details.

The key tool used to prove Theorem 1, which appears throughout the technical portions of this paper, is the *vector-valued Rademacher complexity* due to [30].

**Definition 3.** *For a function class $\mathcal{G} \subseteq (\mathcal{Z} \to \mathbb{R}^K)$, the vector-valued Rademacher complexity is*

$$\overrightarrow{\mathfrak{R}}(g; z_{1:n}) := \mathbb{E}_{\boldsymbol{\epsilon}} \sup_{g \in \mathcal{G}} \sum_{t=1}^n \langle \boldsymbol{\epsilon}_t, g(z_t) \rangle. \tag{5}$$

The vector-valued Rademacher complexity arises through an elegant contraction trick due to Maurer.

**Theorem 2** (Vector-valued contraction [30]). *Let* $\mathcal{G} \subseteq (\mathcal{Z} \to \mathbb{R}^K)$, *and let* $h_t : \mathbb{R}^K \to \mathbb{R}$ *be a sequence of functions for* $t \in [n]$, *each of which is L-Lipschitz with respect to* $\ell_2$. *Then*

$$\mathbb{E}_\epsilon \sup_{g \in \mathcal{G}} \sum_{t=1}^n \epsilon_t h_t(g(z_t)) \leq \sqrt{2}L \cdot \vec{\mathfrak{R}}(\mathcal{G}\,;z_{1:n}). \tag{6}$$

We remark that while our applications require only gradient uniform convergence, we anticipate that the tools of this section will find use in settings where convergence of higher-order derivatives is needed to ensure success of optimization routines. To this end, we have extended the chain rule (Theorem 1) to handle Hessian convergence; see Appendix E.

## 3 Application: Smooth Models

In this section we instantiate the general gradient uniform convergence tools and the GD condition to derive optimization consequences for two standard settings previously studied by [31]: generalized linear models and robust regression.

**Generalized Linear Model**   We first consider the problem of learning a generalized linear model with the square loss.[1] Fix a norm $\|\cdot\|$, take $\mathcal{X} = \{x \in \mathbb{R}^d \mid \|x\| \leq R\}$, $\mathcal{W} = \{w \in \mathbb{R}^d \mid \|w\|_\star \leq B\}$, and $\mathcal{Y} = \{0,1\}$. Choose a link function $\sigma : \mathbb{R} \to [0,1]$ and define the loss to be $\ell(w\,;x,y) = (\sigma(\langle w,x \rangle) - y)^2$. Standard choices for $\sigma$ include the logistic link function $\sigma(s) = (1 + e^{-s})^{-1}$ and the probit link function $\sigma(s) = \Phi(s)$, where $\Phi$ is the gaussian cumulative distribution function.

To establish the GD property and provide uniform convergence bounds, we make the following regularity assumptions on the loss.

**Assumption 1** (Generalized Linear Model Regularity). Let $\mathcal{S} = [-BR, BR]$.

   (a) $\exists C_\sigma \geq 1$ s.t. $\max\{\sigma'(s), \sigma''(s)\} \leq C_\sigma$ for all $s \in \mathcal{S}$.

   (b) $\exists c_\sigma > 0$ s.t. $\sigma'(s) \geq c_\sigma$ for all $s \in \mathcal{S}$.

   (c) $\mathbb{E}[y \mid x] = \sigma(\langle w^\star, x \rangle)$ for some $w^\star \in \mathcal{W}$.

Assumption (a) suffices to bound the normed Rademacher complexity $\mathfrak{R}_{\|\cdot\|}(\nabla\ell \circ \mathcal{W})$, and combined with (b) and (c) the assumption implies that $L_\mathcal{D}$ satisfies three variants of GD condition, and this leads to three final rates: a dimension-independent "slow rate" that holds for any smooth norm, a dimension-dependent fast rate for the $\ell_2$ norm, and a sparsity-dependent fast rate that holds under an additional restricted eigenvalue assumption. This gives rise to a family of generic excess risk bounds.

To be precise, let us introduce some additional notation: $\Sigma = \mathbb{E}_x[xx^\top]$ is the data covariance matrix and $\lambda_{\min}(\Sigma)$ denotes the minimum non-zero eigenvalue. For sparsity dependent fast rates, define $\mathcal{C}(S, \alpha) := \{\nu \in \mathbb{R}^d \mid \|\nu_{S^C}\|_1 \leq \alpha\|\nu_S\|_1\}$ and let $\psi_{\min}(\Sigma) = \inf_{\nu \in \mathcal{C}(S(w^\star),1)} \frac{\langle \nu, \Sigma\nu \rangle}{\langle \nu, \nu \rangle}$ be the restricted eigenvalue.[2] Lastly, recall that a norm $\|\cdot\|$ is said to be $\beta$-smooth if the function $\Psi(x) = \frac{1}{2}\|x\|^2$ has $\beta$-Lipschitz gradients with respect to $\|\cdot\|$.

**Theorem 3.** *For the generalized linear model setting, the following excess risk inequalities each hold with probability at least* $1 - \delta$ *over the draw of the data* $\{(x_t, y_t)\}_{t=1}^n$ *for any algorithm* $\widehat{w}^{\mathrm{alg}}$:
• **Norm-Based/High-Dimensional Setup.**   *When* $\mathcal{X}$ *is the ball for* $\beta$-*smooth norm* $\|\cdot\|$ *and* $\mathcal{W}$ *is the dual ball,*

$$L_\mathcal{D}(\widehat{w}^{\mathrm{alg}}) - L^\star \leq \mu_h \cdot \left\|\nabla\widehat{L}_n(\widehat{w}^{\mathrm{alg}})\right\| + \frac{C_h}{\sqrt{n}}.$$

• **Low-Dimensional** $\ell_2/\ell_2$ **Setup.**   *When* $\mathcal{X}$ *and* $\mathcal{W}$ *are both* $\ell_2$ *balls:*

$$L_\mathcal{D}(\widehat{w}^{\mathrm{alg}}) - L^\star \leq \frac{1}{\lambda_{\min}(\Sigma)}\left(\mu_l \cdot \left\|\nabla\widehat{L}_n(\widehat{w}^{\mathrm{alg}})\right\|^2 + \frac{C_l}{n}\right).$$

| Model | Algorithm | Sample Complexity | |
|---|---|---|---|
| | | Norm-based/Infinite dim. | Low-dim. |
| Generalized Linear | Proposition 3 | $O(\varepsilon^{-2})$ | $O(d\varepsilon^{-1})$ |
| | [31] Theorem 4 | n/a | $O(d\varepsilon^{-1})$ |
| | GLMtron [21] | $O(\varepsilon^{-2})$ | n/a |
| Robust Regression | Proposition 3 | $O(\varepsilon^{-2})$ | $O(d\varepsilon^{-1})$ |
| | [31] Theorem 6 | n/a | $O(d\varepsilon^{-1})$ |

Table 1: Sample complexity comparison. Highlighted cells indicate optimal sample complexity.

- **Sparse $\ell_\infty/\ell_1$ Setup.** When $\mathcal{X}$ is the $\ell_\infty$ ball, $\mathcal{W}$ is the $\ell_1$ ball, and $\|w^\star\|_1 = B$:[3]

$$L_\mathcal{D}(\widehat{w}^{\mathrm{alg}}) - L^\star \leq \frac{\|w^\star\|_0}{\psi_{\min}(\Sigma)}\left(\mu_s \cdot \left\|\nabla\widehat{L}_n(\widehat{w}^{\mathrm{alg}})\right\|^2 + \frac{C_s}{n}\right).$$

*The quantities $C_h/C_l/C_s$ and $\mu_h/\mu_l/\mu_s$ are constants depending on $(B, R, C_\sigma, c_\sigma, \beta, \log(\delta^{-1}))$ but not explicitly on the dimension $d$ (beyond logarithmic factors) or complexity of the class $\mathcal{W}$ (beyond $B$ and $R$).*

Precise statements for the problem dependent constants in Theorem 3 including dependence on the norms $R$ and $B$ can be found in Appendix C.

We now formally introduce the robust regression setting and provide a similar guarantee.

**Robust Regression** Fix a norm $\|\cdot\|$ and take $\mathcal{X} = \{x \in \mathbb{R}^d \mid \|x\| \leq R\}$, $\mathcal{W} = \{w \in \mathbb{R}^d \mid \|w\|_\star \leq B\}$, and $\mathcal{Y} = [-Y, Y]$ for some constant $Y$. We pick a potentially non-convex function $\rho : \mathbb{R} \to \mathbb{R}_+$ and define the loss via $\ell(w; x, y) = \rho(\langle w, x\rangle - y)$. Non-convex choices for $\rho$ arise in robust statistics, with a canonical example being Tukey's biweight loss.[4] While optimization is clearly not possible for arbitrary choices of $\rho$, the following assumption is sufficient to guarantee that the population risk $L_\mathcal{D}$ satisfies the GD property.

**Assumption 2** (Robust Regression Regularity). Let $\mathcal{S} = [-(BR + Y), (BR + Y)]$.

(a) $\exists C_\rho \geq 1$ s.t. $\max\{\rho'(s), \rho''(s)\} \leq C_\rho$ for all $s \in \mathcal{S}$.

(b) $\rho'$ is odd with $\rho'(s) > 0$ for all $s > 0$ and $h(s) := \mathbb{E}_\zeta[\rho'(s + \zeta)]$ has $h'(0) > c_\rho$.

(c) There is $w^\star \in \mathcal{W}$ such that $y = \langle w^\star, x\rangle + \zeta$, and $\zeta$ is symmetric zero-mean given $x$.

Similar to the generalized linear model setup, the robust regression setup satisfies three variants of the GD depending on assumptions on the norm $\|\cdot\|$ and the data distribution.

**Theorem 4.** *For the robust regression setting, the following excess risk inequalities each hold with probability at least $1 - \delta$ over the draw of the data $\{(x_t, y_t)\}_{t=1}^n$ for any algorithm $\widehat{w}^{\mathrm{alg}}$:*
- **Norm-Based/High-Dimensional Setup.** *When $\mathcal{X}$ is the ball for $\beta$-smooth norm $\|\cdot\|$ and $\mathcal{W}$ is the dual ball,*

$$L_\mathcal{D}(\widehat{w}^{\mathrm{alg}}) - L^\star \leq \mu_h \cdot \left\|\nabla\widehat{L}_n(\widehat{w}^{\mathrm{alg}})\right\| + \frac{C_h}{\sqrt{n}}.$$

- **Low-Dimensional $\ell_2/\ell_2$ Setup.** *When $\mathcal{X}$ and $\mathcal{W}$ are both $\ell_2$ balls:*

$$L_\mathcal{D}(\widehat{w}^{\mathrm{alg}}) - L^\star \leq \frac{1}{\lambda_{\min}(\Sigma)}\left(\mu_l \cdot \left\|\nabla\widehat{L}_n(\widehat{w}^{\mathrm{alg}})\right\|^2 + \frac{C_l}{n}\right).$$

- **Sparse $\ell_\infty/\ell_1$ Setup.** *When $\mathcal{X}$ is the $\ell_\infty$ ball, $\mathcal{W}$ is the $\ell_1$ ball, and $\|w^\star\|_1 = B$:*

$$L_\mathcal{D}(\widehat{w}^{\mathrm{alg}}) - L^\star \leq \frac{\|w^\star\|_0}{\psi_{\min}(\Sigma)}\left(\mu_s \cdot \left\|\nabla\widehat{L}_n(\widehat{w}^{\mathrm{alg}})\right\|^2 + \frac{C_s}{n}\right).$$

*The constants $C_h/C_l/C_s$ and $\mu_h/\mu_l/\mu_s$ depend on $(B, R, C_\rho, c_\rho, \beta, \log(\delta^{-1}))$, but not explicitly on the dimension $d$ (beyond $\log$ factors) or complexity of the class $\mathcal{W}$ (beyond the range parameters $B$ and $R$).*

Theorem 3 and Theorem 4 immediately imply that standard non-convex *optimization* algorithms for finding stationary points can be converted to non-convex *learning* algorithms with optimal sample complexity; this is summarized by the following theorem, focusing on the "high-dimensional" and "low-dimensional" setups above in the case of the $\ell_2$ norm for simplicity.

**Proposition 3.** Suppose that $\Sigma \geq \frac{1}{d}I$, $\|\cdot\| = \ell_2$. Consider the following meta-algorithm for the non-convex generalized linear model (under Assumption 1) and robust regression (under Assumption 2) setting.

1. Gather $n = \frac{1}{\varepsilon^2} \wedge \frac{d}{\varepsilon}$ samples $\{(x_t, y_t)\}_{t=1}^n$.

2. Find a point $\widehat{w}^{\mathrm{alg}} \in \mathcal{W}$ with $\left\| \nabla \widehat{L}_n(\widehat{w}^{\mathrm{alg}}) \right\| \leq O\left(\frac{1}{\sqrt{n}}\right)$, which is guaranteed to exist.

This meta-algorithm guarantees $\mathbb{E} L_{\mathcal{D}}(\widehat{w}^{\mathrm{alg}}) - L^\star \leq C \cdot \varepsilon$, where $C$ is a problem-dependent but dimension-independent[5] constant.

There are many non-convex optimization algorithms that provably find an approximate stationary point of the empirical risk, including gradient descent [33], SGD [10], and Non-convex SVRG [38, 3]. Note, however, that these algorithms are not generically guaranteed to satisfy the constraint $\widehat{w}^{\mathrm{alg}} \in \mathcal{W}$ a-priori. We can circumvent this difficulty and take advantage of these generic algorithms by instead finding stationary points of the *regularized* empirical risk. We show that any algorithm that finds a (unconstrained) stationary point of the regularized empirical risk indeed the obtains optimal $O\left(\frac{1}{\varepsilon^2}\right)$ sample complexity in the norm-based regime.

**Theorem 9** (informal). *Suppose we are in the generalized linear model setting or robust regression setting. Let $\widehat{L}_n^\lambda(w) = \widehat{L}_n(w) + \frac{\lambda}{2}\|w\|_2^2$. For any $\delta > 0$ there is a setting of the regularization parameter $\lambda$ such that any $\widehat{w}^{\mathrm{alg}}$ with $\nabla \widehat{L}_n^\lambda(\widehat{w}^{\mathrm{alg}}) = 0$ guarantees $L_{\mathcal{D}}(\widehat{w}^{\mathrm{alg}}) - L^\star \leq \tilde{O}\left(\sqrt{\frac{\log(\delta^{-1})}{n}}\right)$ with probability at least $1 - \delta$.*

See Appendix C.3 for the full theorem statement and proof.

Now is a good time to discuss connections to existing work in more detail.

a) The sample complexity $O\left(\frac{1}{\varepsilon^2} \wedge \frac{d}{\varepsilon}\right)$ for Proposition 3 is optimal up to dependence on Lipschitz constants and the range parameters $B$ and $R$ [42]. The "high-dimensional" $O\left(\frac{1}{\varepsilon^2}\right)$ regime is particularly interesting, and goes beyond recent analyses to non-convex statistical learning [31], which use arguments involving pointwise covers of the space $\mathcal{W}$ and thus have unavoidable dimension-dependence. This highlights the power of the norm-based complexity analysis.

b) In the low-dimensional $O\left(\frac{d}{\varepsilon}\right)$ sample complexity regime, Theorem 3 and Theorem 4 recovers the rates of [31] under the same assumptions—see Appendix C.3 for details. Notably, this is the case even when the radius $R$ is not constant. Note however that when $B$ and $R$ are large the constants in Theorem 3 and Theorem 4 can be quite poor. For the logistic link it is only possible to guarantee $c_\sigma \geq e^{-BR}$, and so it may be more realistic to assume $BR$ is constant.

c) The GLMtron algorithm of [21] also obtains $O\left(\frac{1}{\varepsilon^2}\right)$ for the GLM setting. Our analysis shows that this sample complexity does not require specialized algorithms; any first-order stationary point finding algorithm will do. GLMtron has no guarantees in the $O\left(\frac{d}{\varepsilon}\right)$ regime, whereas our meta-algorithm works in both high- and low-dimensional regimes. A significant benefit of GLMtron, however, is that it does not require a lower bound on the derivative of the link function $\sigma$. It is not clear if this assumption can be removed from our analysis.

d) As an alternative approach, stochastic optimization methods for finding first-order stationary points can be used to directly find an approximate stationary point of the population risk

$\|L_{\mathcal{D}}(w)\| \le \varepsilon$, so long as they draw a fresh sample at each step. In the high-dimensional regime it is possible to show that stochastic gradient descent (and for general smooth norms, mirror descent) obtains $O\left(\frac{1}{\varepsilon^2}\right)$ sample complexity through this approach; this is sketched in Appendix C.3. This approach relies on returning a randomly selected iterate from the sequence and only gives an in-expectation sample complexity guarantee, whereas Theorem 9 gives a high-probability guarantee.

Also, note that many stochastic optimization methods can exploit the $(2, \mu)$-GD condition. Suppose we are in the low-dimensional regime with $\Sigma \succeq \frac{1}{d}I$. The fastest GD-based stochastic optimization method that we are aware of is SNVRG [45], which under the $(2, O(d))$-GD condition will obtain $\varepsilon$ excess risk with $O\left(\frac{d}{\varepsilon} + \frac{d^{3/2}}{\varepsilon^{1/2}}\right)$ sample complexity.

This discussion is summarized in Table 3.

## 4 Non-Smooth Models

In the previous section we used gradient uniform convergence to derive immediate optimization and generalization consequences by finding approximate stationary points of smooth non-convex functions. In practice—notably in deep learning—it is common to optimize *non-smooth* non-convex functions; deep neural networks with rectified linear units (ReLUs) are the canonical example [24, 16]. In theory, it is trivial to construct non-smooth functions for which finding approximate stationary points is intractable (see discussion in [2]), but it appears that in practice stochastic gradient descent can indeed find approximate stationary points of the empirical loss in standard neural network architectures [43]. It is desirable to understand whether gradient generalization can occur in this setting.

The first result of this section is a lower bound showing that even for the simplest possible non-smooth model—a single ReLU—it is impossible to achieve dimension-independent uniform convergence results similar to those of the previous section. On the positive side, we show that it *is* possible to obtain dimension-independent rates under an additional margin assumption.

The full setting is as follows: $\mathcal{X} \subseteq \left\{x \in \mathbb{R}^d \mid \|x\|_2 \le 1\right\}$, $\mathcal{W} \subseteq \left\{w \in \mathbb{R}^d \mid \|w\|_2 \le 1\right\}$, $\mathcal{Y} = \{-1, +1\}$, and $\ell(w\,;x,y) = \sigma(-\langle w, x\rangle \cdot y)$, where $\sigma(s) = \max\{s, 0\}$; this essentially matches the classical Perceptron setup. Note that the loss is not smooth, and so the gradient is not well-defined everywhere. Thus, to make the problem well-defined, we consider convergence for the following representative from the subgradient: $\nabla \ell(w\,;x,y) \coloneqq -y\mathbb{1}\{y\langle w, x\rangle \le 0\} \cdot x$.[6] Our first theorem shows that gradient uniform convergence for this setup must depend on dimension, even when the weight norm $B$ and data norm $R$ are held constant.

**Theorem 5.** *Under the problem setting defined above, for all $n \in \mathbb{N}$ there exist a sequence of instances $\{(x_t, y_t)\}_{t=1}^n$ such that*

$$\mathbb{E}_\epsilon \sup_{w \in \mathcal{W}} \left\| \sum_{t=1}^n \epsilon_t \nabla \ell(w\,;x_t, y_t) \right\|_2 = \Omega\left(\sqrt{dn} \wedge n\right).$$

This result contrasts the setting where $\sigma$ is smooth, where the techniques from Section 2 easily yield a dimension-independent $O(\sqrt{n})$ upper bound on the Rademacher complexity. This is perhaps not surprising since the gradients are discrete functions of $w$, and indeed VC-style arguments suffice to establish the lower bound.

In the classical statistical learning setting, the main route to overcoming dimension dependence—e.g., for linear classifiers—is to assume a *margin*, which allows one to move from a discrete class to a real-valued class upon which a dimension-independent Rademacher complexity bound can be applied [39]. Such arguments have recently been used to derive dimension-independent function value uniform convergence bounds for deep ReLU networks as well [6, 11]. However, this analysis relies on one-sided control of the loss, so it is not clear whether it extends to the inherently directional problem of gradient convergence. Our main contribution in this section is to introduce additional machinery to prove dimension-free gradient convergence under a new type of margin assumption.

**Definition 4.** *Given a distribution $P$ over the support $\mathcal{X}$ and an increasing function $\phi : [0,1] \to [0,1]$, any $w \in \mathcal{W}$ is said to satisfy the $\phi$-**soft-margin condition with respect to** $P$ if*

$$\forall \gamma \in [0,1], \quad \mathbb{E}_{x \sim P}\Big[\mathbb{1}\Big\{\tfrac{|\langle w,x\rangle|}{\|w\|_2 \|x\|_2} \le \gamma\Big\}\Big] \le \phi(\gamma). \tag{7}$$

We call $\phi$ a *margin function*. We define the set of all weights that satisfy the $\phi$-soft-margin condition with respect to a distribution $P$ via:

$$\mathcal{W}(\phi, P) = \Big\{ w \in \mathcal{W} \ : \ \forall \gamma \in [0,1], \ \mathbb{E}_{x \sim P}\Big[\mathbb{1}\Big\{\tfrac{|\langle w,x\rangle|}{\|w\|_2 \|x\|_2} \le \gamma\Big\}\Big] \le \phi(\gamma)\Big\}. \tag{8}$$

Of particular interest is $\mathcal{W}(\phi, \widehat{\mathcal{D}}_n)$, the set of all the weights that satisfy the $\phi$-soft-margin condition with respect to the empirical data distribution. That is, any $w \in \mathcal{W}(\phi, \widehat{\mathcal{D}}_n)$ predicts with at least a $\gamma$ margin on all but a $\phi(\gamma)$ fraction of the data. The following theorem provides a dimension-independent uniform convergence bound for the gradients over the class $\mathcal{W}(\phi, \widehat{\mathcal{D}}_n)$ for any margin function $\phi$ fixed in advance.

**Theorem 6.** *Let $\phi : [0,1] \to [0,1]$ be a fixed margin function. With probability at least $1 - \delta$ over the draw of the data $\{(x_t, y_t)\}_{t=1}^n$,*

$$\sup_{w \in \mathcal{W}(\phi, \widehat{\mathcal{D}}_n)} \big\| \nabla L_{\mathcal{D}}(w) - \nabla \widehat{L}_n(w) \big\|_2 \le \tilde{O}\left( \inf_{\gamma > 0} \left\{ \sqrt{\phi(4\gamma)} + \frac{1}{\gamma} \sqrt{\frac{\log\left(\frac{1}{\delta}\right)}{n}} + \frac{1}{\gamma^{\frac{1}{2}} n^{\frac{1}{4}}} \right\} \right),$$

*where $\tilde{O}(\cdot)$ hides $\log\log(\frac{1}{\gamma})$ and $\log n$ factors.*

As a concrete example, when $\phi(\gamma) = \gamma^{\frac{1}{2}}$ Theorem 6 yields a dimension-independent uniform convergence bound of $O(n^{-\frac{1}{12}})$, thus circumventing the lower bound of Theorem 5 for large values of $d$.

## 5   Discussion

We showed that vector Rademacher complexities are a simple and effective tool for deriving dimension-independent uniform convergence bounds and used these bounds in conjunction with the (population) Gradient Domination property to derive optimal algorithms for non-convex statistical learning in high and infinite dimension. We hope that these tools will find broader use for norm-based capacity control in non-convex learning settings beyond those considered here. Of particular interest are models where convergence of higher-order derivatives is needed to ensure success of optimization routines. Appendix E contains an extension of Theorem 1 for Hessian uniform convergence, which we anticipate will find use in such settings.

In Section 3 we analyzed generalized linear models and robust regression using both the $(1, \mu)$-GD property and the $(2, \mu)$-GD property. In particular, the $(1, \mu)$-GD property was critical to obtain dimension-independent norm-based capacity control. While there are many examples of models for which the population risk satisfies $(2, \mu)$-GD property (phase retrieval [40, 44], ResNets with linear activations [13], matrix factorization [29], blind deconvolution [27]), we do not know whether the $(1, \mu)$-GD property holds for these models. Establishing this property and consequently deriving dimension-independent optimization guarantees is an exciting future direction.

Lastly, an important question is to analyze non-smooth problems beyond the simple ReLU example considered in Section 4. See [9] for subsequent work in this direction.

**Acknowledgements**   K.S acknowledges support from the NSF under grants CDS&E-MSS 1521544 and NSF CAREER Award 1750575, and the support of an Alfred P. Sloan Fellowship. D.F. acknowledges support from the NDSEG PhD fellowship and Facebook PhD fellowship.

## Footnotes

[1][31] refer to the model as "binary classification", since $\sigma$ can model conditional probabilities of two classes.

[2]Recall that $S(w^\star) \subseteq [d]$ is the set of non-zero entries of $w^\star$, and for any vector $w$, $w_S \in \mathbb{R}^d$ refers to the vector $w$ with all entries in $S^C$ set to zero (as in [37]).

[3]The constraint $\|w^\star\|_1 = B$ simplifies analysis of generic algorithms in the vein of constrained LASSO [41].

[4]For a fixed parameter $c > 0$ the biweight loss is defined via $\rho(t) = \frac{c^2}{6} \cdot \begin{cases} 1 - (1 - (t/c)^2)^3, & |t| \leq c. \\ 1, & |t| \geq c. \end{cases}$

[5]Whenever $B$ and $R$ are constant.

[6]For general non-convex and non-smooth functions one can extend this approach by considering convergence for a representative from the Clarke sub-differential [7, 8].

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
