[Supplementary Material]

# A  Preliminaries

## A.1  Contraction Lemmas

**Lemma 1** (e.g. [25])**.** Let $\mathcal{F}$ be any scalar-valued function class and $\phi_1, \dots, \phi_n$ be any sequence of functions where $\phi_t : \mathbb{R} \to \mathbb{R}$ is $L$-Lipschitz. Then

$$\mathbb{E}_\epsilon \sup_{f \in \mathcal{F}} \sum_{t=1}^n \epsilon_t \phi_t(f(x_t)) \le L \cdot \mathbb{E}_\epsilon \sup_{f \in \mathcal{F}} \sum_{t=1}^n \epsilon_t f(x_t). \tag{9}$$

The following is a weighted generalization of the vector-valued Lipschitz contraction inequality.

**Lemma 2.** Let $\mathcal{F} \subseteq (\mathcal{X} \to \mathbb{R}^K)$, and let $h_t : \mathbb{R}^K \to \mathbb{R}$ be a sequence of functions for $t \in [n]$. Suppose each $h_t$ is 1-Lipschitz with respect to $\|z\|_{A_t} := \sqrt{\langle z, A_t z \rangle}$, where $A_t \in \mathbb{R}^{K \times K}$ is positive semidefinite. Then

$$\mathbb{E}_\epsilon \sup_{f \in \mathcal{F}} \sum_{t=1}^n \epsilon_t h_t(f(x_t)) \le \sqrt{2} \, \mathbb{E}_\epsilon \sup_{f \in \mathcal{F}} \sum_{t=1}^n \Big\langle \epsilon_t, A_t^{1/2} f(x_t) \Big\rangle. \tag{10}$$

**Proof sketch for Lemma 2.** Same proof as Theorem 3 in [30], with the additional observation that $\|z\|_{A_t} = \left\| A_t^{1/2} z \right\|_2$. $\qquad\square$

**Lemma 3.** Let $\mathcal{G}$ be a class of vector-valued functions whose output space forms $M$ blocks of vectors, i.e. each $g \in \mathcal{G}$ has the form $g : \mathcal{Z} \to \mathbb{R}^{d_1 + d_2 + \cdots + d_M}$, where $g(z)_i \in \mathbb{R}^{d_i}$ denotes the $i$th block. Let $h_t : \mathbb{R}^{d_1 + d_2 + \cdots + d_M} \to \mathbb{R}$, be a sequence of functions for $t \in [n]$ that satisfy the following block-wise Lipschitz property: For any assignment $a_1, \dots, a_M$ with each $a_i \in \mathbb{R}^{d_i}$, $h_t(a_1, \dots, a_M)$ is $L_i$-Lipschitz with respect to $a_i$ in the $\ell_2$ norm. Then

$$\mathbb{E}_\epsilon \sup_{g \in \mathcal{G}} \sum_{t=1}^n h_t(g_1(z_t), \dots, g_M(z_t)) \le \sqrt{2M} \sum_{i=1}^M L_i \, \mathbb{E}_\epsilon \sup_{f \in \mathcal{F}} \sum_{t=1}^n \langle \epsilon_t, g_i(z_t) \rangle.$$

**Proof.** Immediate consequence of Lemma 2, along with sub-additivity of the supremum. $\qquad\square$

## A.2  Bound for Vector-Valued Random Variables

**Definition 5.** *For any vector space $\mathcal{V}$, a convex function $\Psi : \mathcal{V} \to \mathbb{R}$ is $\beta$-smooth with respect to a norm $\|\cdot\|$ if*

$$\Psi(x) \le \Psi(y) + \langle \nabla \Psi(y), x - y \rangle + \frac{\beta}{2} \|x - y\|^2 \quad \forall x, y \in \mathcal{V}.$$

*A norm $\|\cdot\|$ is said to be $\beta$-smooth if the function $\Psi(x) = \frac{1}{2} \|x\|^2$ is $\beta$-smooth with respect to $\|\cdot\|$.*

**Theorem 7.** *Let $\|\cdot\|$ be any norm for which there exists $\Psi$ such that $\Psi(x) \ge \frac{1}{2} \|x\|^2$, $\Psi(0) = 0$, and $\Psi$ is $\beta$-smooth with respect to $\|\cdot\|$. Then*

$$\mathbb{E}_\epsilon \left\| \sum_{t=1}^n \epsilon_t x_t \right\| \le \sqrt{\beta \sum_{t=1}^n \|x_t\|^2}.$$

The reader may consult [34] for a high-probability version of this theorem.

**Fact 1.** The following spaces and norms satisfy the preconditions of Theorem 7:

- $(\mathbb{R}^d, \ell_p)$ for $p \ge 2$, with $\beta = p - 1$ [19].

- $(\mathbb{R}^d, \ell_\infty)$, with $\beta = O(\log d)$ [19].

- $(\mathbb{R}^{d_1 \times d_2}, \|\cdot\|_\sigma)$, with $\beta = O(\log(d_1 + d_2))$ [22].

**Proof of Theorem 7.** Using Jensen's inequality and the upper bound property of $\Psi$ we have

$$\mathbb{E}_\epsilon \left\| \sum_{t=1}^n \epsilon_t x_t \right\| \le \sqrt{\mathbb{E}_\epsilon \left\| \sum_{t=1}^n \epsilon_t x_t \right\|^2} \le \sqrt{2} \cdot \sqrt{\mathbb{E}_\epsilon \Psi\left( \sum_{t=1}^n \epsilon_t x_t \right)}.$$

Applying the smoothness property at time $n$, and using that $\epsilon_n$ is independent of $\epsilon_1, \ldots, \epsilon_{n-1}$:

$$\sqrt{\mathbb{E}_\epsilon \, \Psi\left(\sum_{t=1}^{n} \epsilon_t x_t\right)} \leq \sqrt{\mathbb{E}_\epsilon\left[\Psi\left(\sum_{t=1}^{n-1} \epsilon_t x_t\right) + \left\langle \Psi\left(\sum_{t=1}^{n-1} \epsilon_t x_t\right), \epsilon_n x_n\right\rangle + \frac{\beta}{2}\|x_n\|^2\right]} = \sqrt{\mathbb{E}_\epsilon \, \Psi\left(\sum_{t=1}^{n-1} \epsilon_t x_t\right) + \frac{\beta}{2}\|x_n\|^2}.$$

The result follows by repeating this argument from time $t = n - 1$ to $t = 1$. $\qquad\square$

## B    Proofs from Section 2

**Theorem 8** ([5], Theorem A.2/Lemma A.5)**.** *Let $\mathcal{F} \subseteq (\mathcal{Z} \to \mathbb{R})$ be a class of functions. Let $Z_1, \ldots, Z_n \sim \mathcal{D}$ i.i.d. for some distribution $\mathcal{D}$. Then with probability at least $1 - \delta$ over the draw of $Z_{1:n}$,*

$$\mathbb{E}\sup_{f\in\mathcal{F}}\left|\mathbb{E}_Z f(Z) - \frac{1}{n}\sum_{t=1}^{n} f(Z_t)\right| \leq 4\,\mathbb{E}_\epsilon \sup_{f\in\mathcal{F}} \frac{1}{n}\sum_{t=1}^{n}\epsilon_t f(Z_t) + 4\sup_{f\in\mathcal{F}}\sup_{z\in\mathcal{Z}}|f(Z)| \cdot \frac{\log\left(\frac{2}{\delta}\right)}{n}. \tag{11}$$

**Lemma 4** (Uniform convergence for vector-valued functions)**.** Let $\mathcal{G} \subseteq \{g : \mathcal{Z} \to \mathfrak{B}\}$ for arbitrary set $\mathcal{Z}$ and vector space $\mathfrak{B}$. Let $Z_1, \ldots, Z_n \sim \mathcal{D}$ i.i.d. for some distribution $\mathcal{D}$. Let a norm $\|\cdot\|$ over $\mathfrak{B}$ be fixed. Then with probability at least $1 - \delta$ over the draw of $Z_{1:n}$,

$$\mathbb{E}\sup_{g\in\mathcal{G}}\left\|\mathbb{E}_Z g(Z) - \frac{1}{n}\sum_{t=1}^{n} g(Z_t)\right\| \leq 4\,\mathbb{E}_\epsilon \sup_{g\in\mathcal{G}}\left\|\frac{1}{n}\sum_{t=1}^{n}\epsilon_t g(Z_t)\right\| + 4\sup_{g\in\mathcal{G}}\sup_{Z\in\mathcal{Z}}\|g(Z)\| \cdot \frac{\log\left(\frac{2}{\delta}\right)}{n} \tag{12}$$

for some absolute constant $c > 0$.

**Proof of Lemma 4.** This follows immediately by applying Theorem 8 to the expanded function class $\mathcal{F} := \{Z \mapsto \langle g(Z), v\rangle \mid g \in \mathcal{G}, \|v\|_\star \leq 1\}$. $\qquad\square$

**Proof of Proposition 1.** This is a direct consequence of McDiarmid's inequality. Consider any vector-valued function class of functions $\mathcal{G}$. Let $Z_1, \ldots, Z_n \sim \mathcal{D}$ i.i.d. for some distribution $\mathcal{D}$. Then McDiarmid's inequality implies that with probability at least $1 - \delta$ over the draw of $Z_{1:n}$,

$$\sup_{g\in\mathcal{G}}\left\|\mathbb{E}_Z g(Z) - \frac{1}{n}\sum_{t=1}^{n} g(Z_t)\right\| \leq \mathbb{E}\sup_{g\in\mathcal{G}}\left\|\mathbb{E}_Z g(Z) - \frac{1}{n}\sum_{t=1}^{n} g(Z_t)\right\| + c\cdot\sup_{g\in\mathcal{G}}\sup_{Z\in\mathcal{Z}}\|g(Z)\| \cdot \sqrt{\frac{\log\left(\frac{2}{\delta}\right)}{n}}. \tag{13}$$

$\qquad\square$

**Proof of Proposition 2.** This follows by applying the uniform convergence lemma, Lemma 4, to the class $\mathcal{G} = \{(x, y) \mapsto \nabla\ell(w\,;x,y) \mid w \in \mathcal{W}\}$. $\qquad\square$

**Proof of Theorem 1.** We write

$$\mathbb{E}_\epsilon \sup_{w\in\mathcal{W}}\left\|\sum_{t=1}^{n}\epsilon_t\nabla(G_t(F_t(w)))\right\| = \mathbb{E}_\epsilon \sup_{w\in\mathcal{W}}\sup_{v\in\mathfrak{B}^\star:\|v\|_\star\leq 1}\sum_{t=1}^{n}\epsilon_t\langle\nabla(G_t(F_t(w))), v\rangle,$$

Using the chain rule for differentiation we have

$$\langle\nabla(G_t(F_t(w))), v\rangle = \left\langle(\nabla G_t)(F_t(w)), (\langle\nabla F_{t,k}(w), v\rangle)_{k\in[K]}\right\rangle.$$

We now introduce new functions that relabel the quantities in this expression. Let $h : \mathbb{R}^{2K} \to \mathbb{R}$ be given by $h(a, b) = \langle a, b\rangle$, let $f_1 : \mathcal{W} \to \mathbb{R}^K$ be given by $f_1(w) = (\nabla G_t)(F_t(w))$ and $f_2$ be given by $f_2(w, v) = (\langle\nabla F_{t,k}(w), v\rangle)_{k\in[K]}$. We apply the block-wise contraction lemma Lemma 3 with one block for $f_1$ and one block for $f_2$ to conclude

$$\mathbb{E}_\epsilon \sup_{w\in\mathcal{W}}\sup_{v\in\mathfrak{B}^\star:\|v\|_\star\leq 1}\sum_{t=1}^{n}\epsilon_t h(f_1(w), f_2(w, v))$$

$$\leq 2L_F\,\mathbb{E}_\epsilon \sup_{w\in\mathcal{W}}\sup_{v\in\mathfrak{B}^\star:\|v\|_\star\leq 1}\sum_{t=1}^{n}\langle\boldsymbol{\epsilon}_t, f_1(w)\rangle + 2L_G\,\mathbb{E}_\epsilon \sup_{w\in\mathcal{W}}\sup_{v\in\mathfrak{B}^\star:\|v\|_\star\leq 1}\sum_{t=1}^{n}\langle\boldsymbol{\epsilon}_t, f_2(w, v)\rangle,$$

which establishes the result after expanding terms. All that must be verified is that the assumptions on the norm bounds for $\nabla G_t$ and $\nabla F_t$ in the theorem statement ensure the the Lipschitz requirement in the statement of Lemma 3 is met. $\qquad\square$

# C Proofs from Section 3

For all proofs in this section we adopt the notation $s := \|w^\star\|_0$, and use $c > 0$ to denote an absolute constant whose precise value depends on context.

## C.1 Generalized Linear Models

**Proof of Theorem 3.** To begin, we apply Proposition 1 and Proposition 2 to conclude that whenever $(\alpha, \mu)$-PL holds, with probability at least $1-\delta$ over the examples $\{(x_t, y_t)\}_{t=1}^n$, any learning algorithm $\widehat{w}^{\mathrm{alg}} \in \mathcal{W}$ satisfies

$$L_\mathcal{D}(\widehat{w}^{\mathrm{alg}}) - L^\star \le c \cdot \mu\left(\left\|\nabla\widehat{L}_n(\widehat{w}^{\mathrm{alg}})\right\|^\alpha + \left(\frac{\mathfrak{R}_{\|\cdot\|}(\nabla\ell \circ \mathcal{W}; x_{1:n}, y_{1:n})}{n} + 2C_\sigma R\sqrt{\frac{\log(1/\delta)}{n}}\right)^\alpha\right). \tag{14}$$

Here $c > 0$ is an absolute constant and we have used that $\|\nabla\ell(w; x_t, y_t)\| \le 2C_\sigma R$.

**Smooth high-dimensional setup** For the general smooth norm pair setup in (14), Lemma 5 and Lemma 7 imply

$$L_\mathcal{D}(\widehat{w}^{\mathrm{alg}}) - L^\star \le c \cdot \frac{BC_\sigma}{c_\sigma}\left(\left\|\nabla\widehat{L}_n(\widehat{w}^{\mathrm{alg}})\right\| + \left(BR^2C_\sigma^2\sqrt{\frac{\beta}{n}} + 2C_\sigma R\sqrt{\frac{\log(1/\delta)}{n}}\right)\right)$$

$$= \mu_{\mathrm{h}} \cdot \left\|\nabla\widehat{L}_n(\widehat{w}^{\mathrm{alg}})\right\| + \frac{C_{\mathrm{h}}}{\sqrt{n}}.$$

where we recall $C_{\mathrm{h}} = c \cdot \frac{B^2 R^2 C_\sigma^3 \sqrt{\beta} + 2C_\sigma^2 BR\sqrt{\log(1/\delta)}}{c_\sigma}$ and $\mu_{\mathrm{h}} = c \cdot \frac{BC_\sigma}{c_\sigma}$.

**Low-dimensional $\ell_2/\ell_2$ setup** For the low-dimension $\ell_2/\ell_2$ pair setup in (14), Lemma 5 and Lemma 7 imply

$$L_\mathcal{D}(\widehat{w}^{\mathrm{alg}}) - L^\star \le c \cdot \frac{C_\sigma}{4c_\sigma^3 \lambda_{\min}(\Sigma)}\left(\left\|\nabla\widehat{L}_n(\widehat{w}^{\mathrm{alg}})\right\|^2 + \left(BR^2C_\sigma^2\sqrt{\frac{1}{n}} + 2C_\sigma R\sqrt{\frac{\log(1/\delta)}{n}}\right)^2\right)$$

$$= \frac{\mu_{\mathrm{l}}}{\lambda_{\min}(\Sigma)} \cdot \left\|\nabla\widehat{L}_n(\widehat{w}^{\mathrm{alg}})\right\|^2 + \frac{C_{\mathrm{l}}}{n \cdot \lambda_{\min}(\Sigma)},$$

where we have used that the $\ell_2$ norm is 1-smooth in Lemma 7. Recall that $C_{\mathrm{l}} = c \cdot \frac{2C_\sigma^5 R^4 B^2 + 8C_\sigma^3 R^2 \log(1/\delta)}{4c_\sigma^3}$ and $\mu_{\mathrm{l}} = c \cdot \frac{C_\sigma}{4c_\sigma^3}$.

**Sparse $\ell_\infty/\ell_1$ setup** For the sparse $\ell_\infty/\ell_1$ pair setup in (14), Lemma 5 and Lemma 7 imply

$$L_\mathcal{D}(\widehat{w}^{\mathrm{alg}}) - L^\star \le c \cdot \frac{C_\sigma s}{c_\sigma^3 \psi_{\min}(\Sigma)}\left(\left\|\nabla\widehat{L}_n(\widehat{w}^{\mathrm{alg}})\right\|^2 + \left(BR^2C_\sigma^2\sqrt{\frac{\log d}{n}} + 2C_\sigma R\sqrt{\frac{\log(1/\delta)}{n}}\right)^2\right)$$

$$= \frac{\mu_{\mathrm{s}} \cdot s}{\psi_{\min}(\Sigma)} \cdot \left\|\nabla\widehat{L}_n(\widehat{w}^{\mathrm{alg}})\right\|^2 + \frac{s}{n} \cdot \frac{C_{\mathrm{s}}}{\psi_{\min}(\Sigma)},$$

where we have used that the $\ell_\infty$ norm has the smoothness property with $\beta = O(\log(d))$ in Lemma 7. Recall that $C_{\mathrm{s}} = c \cdot \frac{2C_\sigma^5 R^4 B^2 \log(d) + 8C_\sigma^3 R^2 \log(1/\delta)}{c_\sigma^3}$ and $\mu_{\mathrm{s}} = c \cdot \frac{C_\sigma}{c_\sigma^3}$.

$\square$

**Lemma 5** (GD condition for the GLM). Consider the generalized linear model setup of Section 3.

- When $\|\cdot\|/\|\cdot\|_\star$ are any dual norm pair, we have $\left(1, \frac{BC_\sigma}{c_\sigma}\right)$-GD:

$$L_\mathcal{D}(w) - L_\mathcal{D}(w^\star) \le \frac{BC_\sigma}{c_\sigma}\|\nabla L_\mathcal{D}(w)\| \quad \forall w \in \mathcal{W}. \tag{15}$$

- In the $\ell_2/\ell_2$ setup, we have $\left(2, \frac{C_\sigma}{4c_\sigma^3 \lambda_{\min}(\Sigma)}\right)$-GD:

$$L_\mathcal{D}(w) - L_\mathcal{D}(w^\star) \le \frac{C_\sigma}{4c_\sigma^3 \lambda_{\min}(\Sigma)} \|\nabla L_\mathcal{D}(w)\|_2^2 \quad \forall w \in \mathcal{W}. \tag{16}$$

- In the sparse $\ell_\infty/\ell_1$ setup, where $\|w^\star\|_0 \le s$, we have $\left(2, \frac{C_\sigma s}{c_\sigma^3 \psi_{\min}(\Sigma)}\right)$-GD:

$$L_\mathcal{D}(w) - L_\mathcal{D}(w^\star) \le \frac{C_\sigma s}{c_\sigma^3 \psi_{\min}(\Sigma)} \|\nabla L_\mathcal{D}(w)\|_\infty^2 \quad \forall w \in \mathcal{W}. \tag{17}$$

**Proof of Lemma 5.**
**Upper bound for excess risk.** We first prove the following intermediate upper bound:

$$L_\mathcal{D}(w) - L_\mathcal{D}(w^\star) \le \frac{C_\sigma}{2c_\sigma} \langle \nabla L_\mathcal{D}(w), w - w^\star \rangle. \tag{18}$$

Letting $w \in \mathcal{W}$ be fixed, we have

$$\langle \nabla L_\mathcal{D}(w), w - w^\star \rangle = 2 \, \mathbb{E}_{(x,y)}[(\sigma(\langle w, x \rangle - y)\sigma'(\langle w, x \rangle)\langle w - w^\star, x \rangle].$$

Using the well-specified assumption:

$$= 2 \, \mathbb{E}_x[(\sigma(\langle w, x \rangle - \sigma(\langle w^\star, x \rangle))\sigma'(\langle w, x \rangle)\langle w - w^\star, x \rangle].$$

We now consider the term inside the expectation. Since $\sigma$ is increasing we have

$$\sigma(\langle w, x \rangle - \sigma(\langle w^\star, x \rangle))\sigma'(\langle w, x \rangle)\langle w - w^\star, x \rangle = |\sigma(\langle w, x \rangle - \sigma(\langle w^\star, x \rangle))| \cdot |\langle w - w^\star, x \rangle| \cdot \sigma'(\langle w, x \rangle)$$

point-wise. We apply two lower bounds. First, $\sigma'(\langle w, x \rangle) > c_\sigma$ by assumption. Second, Lipschitzness of $\sigma$ implies

$$|\sigma(\langle w, x \rangle) - \sigma(\langle w^\star, x \rangle)| \le C_\sigma |\langle w - w^\star, x \rangle|.$$

Combining these inequalities, we also obtain the following inequality in expectation over $x$:

$$\mathbb{E}_x(\sigma(\langle w, x \rangle) - \sigma(\langle w^\star, x \rangle))^2 \le \frac{C_\sigma}{2c_\sigma} \langle \nabla L_\mathcal{D}(w), w - w^\star \rangle.$$

Lastly, since the model is well-specified we have

$$L_\mathcal{D}(w) - L_\mathcal{D}(w^\star) = \mathbb{E}_x(\sigma(\langle w, x \rangle) - \sigma(\langle w^\star, x \rangle))^2,$$

by a standard argument:

$$\begin{aligned} L_\mathcal{D}(w) - L_\mathcal{D}(w^\star) &= \mathbb{E}_{x,y}\big[\sigma^2(\langle w, x \rangle) + y^2 - 2\sigma(\langle w, x \rangle)y - \sigma^2(\langle w^\star, x \rangle) - y^2 + 2\sigma(\langle w^\star, x \rangle)y\big] \\ &= \mathbb{E}_x\big[\sigma^2(\langle w, x \rangle) - 2\sigma(\langle w, x \rangle)\sigma(\langle w^\star, x \rangle) + \sigma^2(\langle w^\star, x \rangle)\big] \\ &= \mathbb{E}_x(\sigma(\langle w, x \rangle) - \sigma(\langle w^\star, x \rangle))^2. \end{aligned}$$

**Proving the GD conditions.** With the inequality (18) established the various GD inequalities follow in quick succession.

- $\left(1, \frac{BC_\sigma}{c_\sigma}\right)$-GD:

  To prove this inequality, simply user Hölder's inequality to obtain the upper bound,

  $$\langle \nabla L_\mathcal{D}(w), w - w^\star \rangle \le 2B\|\nabla L_\mathcal{D}(w)\|.$$

- $\left(2, \frac{C_\sigma}{4c_\sigma^3 \lambda_{\min}(\mathbb{E}[xx^\top])}\right)$-GD:

  Resuming from (18) we have

  $$L_\mathcal{D}(w) - L_\mathcal{D}(w^\star) \le \frac{C_\sigma}{2c_\sigma} \langle \nabla L_\mathcal{D}(w), w - w^\star \rangle.$$

Let $P_{\mathcal{X}}$ denote the orthogonal projection onto $\mathrm{span}(\mathbb{E}[xx^\top])$. Note that $\nabla\ell(w\,;x,y)$ is parallel to $x$, we can thus introduce the projection matrix $P_{\mathcal{X}}$ while preserving the inner product

$$= \frac{C_\sigma}{2c_\sigma}\langle P_{\mathcal{X}}\nabla L_{\mathcal{D}}(w), P_{\mathcal{X}}(w - w^\star)\rangle$$

Applying Cauchy-Schwarz:

$$\leq \frac{C_\sigma}{2c_\sigma}\|\nabla L_{\mathcal{D}}(w)\|_2 \cdot \|P_{\mathcal{X}}(w - w^\star)\|_2. \tag{19}$$

What remains is to relate the gradient norm to the term $\|P_{\mathcal{X}}(w - w^\star)\|_2$. We proceed with another lower bound argument similar to the one used to establish (18),

$$\langle \nabla L_{\mathcal{D}}(w), w - w^\star\rangle = 2\,\mathbb{E}_{(x,y)}[(\sigma(\langle w,x\rangle - y)\sigma'(\langle w,x\rangle))\langle w - w^\star, x\rangle].$$

Using the well-specified assumption once more:

$$= 2\,\mathbb{E}_x[(\sigma(\langle w,x\rangle - \sigma(\langle w^\star, x\rangle)))\sigma'(\langle w,x\rangle)\langle w - w^\star, x\rangle].$$

Monotonicity of $\sigma$, implies the argument to the expectation is non-negative pointwise, so we have the lower bound,

$$\geq 2c_\sigma\,\mathbb{E}_x[(\sigma(\langle w,x\rangle - \sigma(\langle w^\star, x\rangle)))\langle w - w^\star, x\rangle].$$

Consider a particular draw of $x$ and assume $\langle w, x\rangle \geq \langle w^\star, x\rangle$ without loss of generality. Using the mean value theorem, there is some $s \in [\langle w^\star, x\rangle, \langle w, x\rangle]$ such that

$$(\sigma(\langle w,x\rangle - \sigma(\langle w^\star, x\rangle)))\langle w - w^\star, x\rangle = \langle w - w^\star, x\rangle^2 \sigma'(s) \geq\ = \langle w - w^\star, x\rangle^2 c_\sigma.$$

Grouping terms, we have shown

$$\langle P_{\mathcal{X}}\nabla L_{\mathcal{D}}(w), P_{\mathcal{X}}(w - w^\star)\rangle = \langle \nabla L_{\mathcal{D}}(w), w - w^\star\rangle \geq 2c_\sigma^2\,\mathbb{E}\langle w - w^\star, x\rangle^2$$
$$= 2c_\sigma^2\langle w - w^\star, \mathbb{E}[xx^\top](w - w^\star)\rangle \tag{20}$$
$$\geq 2c_\sigma^2\lambda_{\min}(\mathbb{E}[xx^\top])\|P_{\mathcal{X}}(w - w^\star)\|_2^2.$$

In other words, by rearranging and applying Cauchy-Schwarz we have

$$\|P_{\mathcal{X}}(w - w^\star)\|_2 \leq \frac{1}{2c_\sigma^2\lambda_{\min}(\mathbb{E}[xx^\top])}\cdot\|\nabla\mathcal{L}_{\mathcal{D}}(w)\|_2.$$

Combining this inequality with (19), we have

$$L_{\mathcal{D}}(w) - L_{\mathcal{D}}(w^\star) \leq \frac{C_\sigma}{4c_\sigma^3\lambda_{\min}(\mathbb{E}[xx^\top])}\cdot\|\nabla\mathcal{L}_{\mathcal{D}}(w)\|_2^2.$$

- $\left(2, \frac{C_\sigma s}{c_\sigma^3\psi_{\min}(\mathbb{E}[xx^\top])}\right)$-GD:

Using the inequality (20) from the preceeding GD proof, we have

$$\langle \nabla L_{\mathcal{D}}(w), w - w^\star\rangle \geq 2c_\sigma^2\langle w - w^\star, \mathbb{E}[xx^\top](w - w^\star)\rangle.$$

By the assumption that $\|w\|_1 \leq \|w^\star\|_1$, we apply Lemma 6 to conclude that 1) $w - w^\star \in \mathcal{C}(S(w^\star), 1)$ and 2) $\|w - w^\star\|_1 \leq 2\sqrt{s}\|w - w^\star\|_2$. The first fact implies that

$$\langle w - w^\star, \mathbb{E}[xx^\top](w - w^\star)\rangle \geq \psi_{\min}(\mathbb{E}[xx^\top])\|w - w^\star\|_2^2.$$

Rearranging, we have

$$\|w - w^\star\|_2 \leq \frac{1}{2c_\sigma^2\psi_{\min}(\mathbb{E}[xx^\top])}\frac{\langle \nabla L_{\mathcal{D}}(w), w - w^\star\rangle}{\|w - w^\star\|_2}$$
$$\leq \frac{1}{2c_\sigma^2\psi_{\min}(\mathbb{E}[xx^\top])}\frac{\|\nabla L_{\mathcal{D}}(w)\|_\infty\|w - w^\star\|_1}{\|w - w^\star\|_2}$$
$$\leq \frac{\sqrt{s}}{c_\sigma^2\psi_{\min}(\mathbb{E}[xx^\top])}\|\nabla L_{\mathcal{D}}(w)\|_\infty.$$

On the other hand, from (18) we have

$$L_{\mathcal{D}}(w) - L_{\mathcal{D}}(w^\star) \le \frac{C_\sigma}{2c_\sigma} \langle \nabla L_{\mathcal{D}}(w), w - w^\star \rangle$$

$$\le \frac{C_\sigma}{2c_\sigma} \|\nabla L_{\mathcal{D}}(w)\|_\infty \|w - w^\star\|_1$$

$$\le \frac{C_\sigma \sqrt{s}}{c_\sigma} \|\nabla L_{\mathcal{D}}(w)\|_\infty \|w - w^\star\|_2.$$

Combining this with the preceding inequality yields the result.

$\square$

The following utility lemma is a standard result in high-dimensional statistics (e.g. [41]).

**Lemma 6.** Let $w, w^\star \in \mathbb{R}^d$. If $\|w\|_1 \le \|w^\star\|_1$ then $w - w^\star =: \nu \in \mathcal{C}(S(w^\star), 1)$. Furthermore, $\|\nu\|_1 \le 2\sqrt{|S(w^\star)|}\|\nu\|_2$.

**Proof of Lemma 6.** Let $S := S(w^\star)$. Then the constraint that $\|w\|_1 \le \|w^\star\|$ implies

$$\|w^\star\|_1 \ge \|w\|_1 = \|w^\star + \nu\|_1 = \|w^\star + \nu_S\|_1 + \|\nu_{S^C}\|_1 \ge \|w^\star\|_1 - \|\nu_S\|_1 + \|\nu_{S^C}\|_1.$$

Rearranging, this implies $\|\nu_{S^C}\|_1 \le \|\nu_S\|_1$, so the first result is established.

For the second result, $\nu \in \mathcal{C}(S, 1)$ implies $\|\nu\|_1 = \|\nu_S\|_1 + \|\nu_{S^C}\|_1 \le 2\|\nu_S\|_1 \le 2\sqrt{|S|}\|\nu_S\|_2 \le 2\sqrt{|S|}\|\nu\|_2$. $\square$

**Lemma 7.** Let the norm $\|\cdot\|$ satisfy the smoothness property of Theorem 7 with constant $\beta$. Then the empirical loss gradient for the generalized linear model setting enjoys the normed Rademacher complexity bound,

$$\mathbb{E}_\epsilon \sup_{w \in \mathcal{W}} \left\| \sum_{t=1}^n \epsilon_t \nabla\ell(w; x_t, y_t) \right\| \le O\left(BR^2 C_\sigma^2 \sqrt{\beta n}\right). \tag{21}$$

**Proof of Lemma 7.** Let $G_t(s) = (\sigma(s) - y_t)^2$ and $F_t(w) = \langle w, x_t \rangle$, so that $\ell(w; x_t, y_t) = G_t(F_t(w))$.

Observe that $G_t'(s) = 2(\sigma(s) - y_t)\sigma'(s)$ and $\nabla F_t(w) = x_t$, so our assumptions imply that that $|G_t'(s)| \le 2C_\sigma$ and $\|\nabla F_t(w)\| \le R$. We can thus apply Theorem 1 to conclude

$$\mathbb{E}_\epsilon \sup_{w \in \mathcal{W}} \left\| \sum_{t=1}^n \epsilon_t \nabla\ell(w; x_t, y_t) \right\| \le 2R \, \mathbb{E}_\epsilon \sup_{w \in \mathcal{W}} \sum_{t=1}^n \epsilon_t G_t'(\langle w, x_t \rangle) + 4C_\sigma \, \mathbb{E}_\epsilon \left\| \sum_{t=1}^n \epsilon_t x_t \right\|.$$

For the first term on the left-hand side, observe that for any $s$, $|G_t''(s)| \le 2|\sigma''(s)| + 2|\sigma'(s)|^2 \le 4C_\sigma^2$, so $G_t'$ is $4C_\sigma^2$-Lipschitz. The classical scalar Lipschitz contraction inequality for Rademacher complexity (Lemma 1) therefore implies

$$\mathbb{E}_\epsilon \sup_{w \in \mathcal{W}} \sum_{t=1}^n \epsilon_t G_t'(\langle w, x_t \rangle) \le 4C_\sigma^2 \, \mathbb{E}_\epsilon \sup_{w \in \mathcal{W}} \sum_{t=1}^n \epsilon_t \langle w, x_t \rangle = 4C_\sigma^2 B \, \mathbb{E}_\epsilon \left\| \sum_{t=1}^n \epsilon_t x_t \right\|.$$

Finally, by our smoothness assumption on the norm, Theorem 7 implies

$$\mathbb{E}_\epsilon \left\| \sum_{t=1}^n \epsilon_t x_t \right\| \le \sqrt{2\beta R^2 n}.$$

$\square$

## C.2 Robust Regression

**Proof of Theorem 4.** This proof follows the same template as Theorem 3. We use Proposition 1 and Proposition 2 to conclude that whenever $(\alpha, \mu)$-PL holds, with probability at least $1 - \delta$ over the examples $\{(x_t, y_t)\}_{t=1}^n$, any learning algorithm $\widehat{w}^{\mathrm{alg}}$ satisfies

$$L_{\mathcal{D}}(\widehat{w}^{\mathrm{alg}}) - L^\star \leq c \cdot \mu \left( \left\| \nabla \widehat{L}_n(\widehat{w}^{\mathrm{alg}}) \right\|^\alpha + \left( \frac{\mathfrak{R}_{\|\cdot\|}(\nabla \ell \circ \mathcal{W}; x_{1:n}, y_{1:n})}{n} + C_\rho R \sqrt{\frac{\log(1/\delta)}{n}} \right)^\alpha \right), \quad (22)$$

where $c > 0$ is an absolute constant and we have used that $\|\nabla \ell(w; x_t, y_t)\| \leq C_\rho R$ with probability 1.

**Smooth high-dimensional setup**   For the general smooth norm pair setup in (22), Lemma 8 and Lemma 9 imply

$$L_{\mathcal{D}}(\widehat{w}^{\mathrm{alg}}) - L^\star \leq c \cdot \frac{B C_\rho}{c_\rho} \left( \left\| \nabla \widehat{L}_n(\widehat{w}^{\mathrm{alg}}) \right\| + \left( B R^2 C_\rho \sqrt{\frac{\beta}{n}} + C_\rho R \sqrt{\frac{\log(1/\delta)}{n}} \right) \right)$$

$$= \mu_{\mathrm{h}} \cdot \left\| \nabla \widehat{L}_n(\widehat{w}^{\mathrm{alg}}) \right\| + \frac{C_{\mathrm{h}}}{\sqrt{n}}.$$

Where we recall $C_{\mathrm{h}} = c \cdot \frac{B^2 R^2 C_\rho^2 \sqrt{\beta} + C_\rho^2 B R \sqrt{\log(1/\delta)}}{c_\rho}$ and $\mu_{\mathrm{h}} = c \cdot \frac{B C_\rho}{c_\rho}$.

**Low-dimensional $\ell_2/\ell_2$ setup**   For the low-dimension $\ell_2/\ell_2$ pair setup in (22), Lemma 8 and Lemma 9 imply

$$L_{\mathcal{D}}(\widehat{w}^{\mathrm{alg}}) - L^\star \leq c \cdot \frac{C_\rho}{2 c_\rho^2 \lambda_{\min}(\Sigma)} \left( \left\| \nabla \widehat{L}_n(\widehat{w}^{\mathrm{alg}}) \right\|^2 + \left( B R^2 C_\rho \sqrt{\frac{1}{n}} + C_\rho R \sqrt{\frac{\log(1/\delta)}{n}} \right)^2 \right)$$

$$= \frac{\mu_{\mathrm{l}}}{\lambda_{\min}(\Sigma)} \cdot \left\| \nabla \widehat{L}_n(\widehat{w}^{\mathrm{alg}}) \right\|^2 + \frac{C_{\mathrm{l}}}{n \cdot \lambda_{\min}(\Sigma)},$$

where we have used that the $\ell_2$ norm is 1-smooth in Lemma 7. Recall that $C_{\mathrm{l}} = c \cdot \frac{C_\rho^3 R^4 B^2 + C_\rho^3 R^2 \log(1/\delta)}{c_\rho^2}$ and $\mu_{\mathrm{l}} = c \cdot \frac{C_\rho}{2 c_\rho^2}$.

**Sparse $\ell_\infty/\ell_1$ setup**   For the sparse $\ell_\infty/\ell_1$ pair setup in (22), Lemma 8 and Lemma 9 imply

$$L_{\mathcal{D}}(\widehat{w}^{\mathrm{alg}}) - L^\star \leq c \cdot \frac{2 C_\rho s}{c_\rho^2 \psi_{\min}(\Sigma)} \left( \left\| \nabla \widehat{L}_n(\widehat{w}^{\mathrm{alg}}) \right\|^2 + \left( B R^2 C_\rho \sqrt{\frac{\log d}{n}} + C_\rho R \sqrt{\frac{\log(1/\delta)}{n}} \right)^2 \right)$$

$$= \frac{\mu_{\mathrm{s}} \cdot s}{\psi_{\min}(\Sigma)} \cdot \left\| \nabla \widehat{L}_n(\widehat{w}^{\mathrm{alg}}) \right\|^2 + \frac{s}{n} \cdot \frac{C_{\mathrm{s}}}{\psi_{\min}(\Sigma)},$$

where we have used that the $\ell_\infty$ norm has the smoothness property with $\beta = O(\log(d))$ in Lemma 7. Recall that $C_{\mathrm{s}} = c \cdot \frac{4 C_\rho^3 R^4 B^2 \log(d) + 4 C_\rho^3 R^2 \log(1/\delta)}{c_\rho^2}$ and $\mu_{\mathrm{s}} = c \cdot \frac{2 C_\rho}{c_\rho^2}$.

$\square$

**Lemma 8** (GD condition for robust regression)**.** Consider the robust regression setup of Section 3.

- When $\|\cdot\|/\|\cdot\|_\star$ are any dual norm pair, we have $\left( 1, \frac{B C_\rho}{c_\rho} \right)$-GD:

$$L_{\mathcal{D}}(w) - L_{\mathcal{D}}(w^\star) \leq \frac{B C_\rho}{c_\rho} \|\nabla L_{\mathcal{D}}(w)\| \quad \forall w \in \mathcal{W}. \quad (23)$$

- In the $\ell_2/\ell_2$ setup, we have $\left(2, \frac{C_\rho}{2c_\rho^2 \lambda_{\min}(\Sigma)}\right)$-GD:

$$L_{\mathcal{D}}(w) - L_{\mathcal{D}}(w^\star) \le \frac{C_\rho}{2c_\rho^2 \lambda_{\min}(\Sigma)} \|\nabla L_{\mathcal{D}}(w)\|_2^2 \quad \forall w \in \mathcal{W}. \tag{24}$$

- In the sparse $\ell_\infty/\ell_1$ setup, where $\|w^\star\|_0 \le s$, we have $\left(2, \frac{2C_\rho s}{c_\rho^2 \psi_{\min}(\Sigma)}\right)$-GD:

$$L_{\mathcal{D}}(w) - L_{\mathcal{D}}(w^\star) \le \frac{2C_\rho s}{c_\rho^2 \psi_{\min}(\Sigma)} \|\nabla L_{\mathcal{D}}(w)\|_\infty^2 \quad \forall w \in \mathcal{W}. \tag{25}$$

**Proof of Lemma 8.**
**Excess risk upper bound.** To begin, smoothness of $\rho$ implies that for any $s, s^\star \in \mathcal{S}$ we have

$$\rho(s) - \rho(s^\star) \le\ \le \rho'(s^\star)(s - s^\star) + \frac{C_\rho}{2}(s - s^\star)^2.$$

Since this holds point-wise, we use it to derive the following in-expectation bound

$$L_{\mathcal{D}}(w) - L_{\mathcal{D}}(w^\star) \le \mathbb{E}_{x,y}[\rho(\langle w^\star, x\rangle - y)\langle w - w^\star, x\rangle] + \frac{C_\rho}{2}\mathbb{E}\langle w - w^\star, x\rangle^2$$

$$= \langle \nabla L_{\mathcal{D}}(w^\star), w - w^\star\rangle + \frac{C_\rho}{2}\mathbb{E}\langle w - w^\star, x\rangle^2.$$

Note however that

$$\nabla L_{\mathcal{D}}(w^\star) = \mathbb{E}_{x,\zeta}[\rho'(-\zeta)x] = 0,$$

since $\zeta$ is conditionally symmetric and $\rho'$ is odd. We therefore have

$$L_{\mathcal{D}}(w) - L_{\mathcal{D}}(w^\star) \le \frac{C_\rho}{2}\mathbb{E}\langle w - w^\star, x\rangle^2.$$

On the other hand, using the form of the gradient we have

$$\langle L_{\mathcal{D}}(w), w - w^\star\rangle = \mathbb{E}_x[\mathbb{E}_\zeta \rho'(\langle w - w^\star, x\rangle - \zeta)\langle w - w^\star, x\rangle]$$

$$= \mathbb{E}_x[h(\langle w - w^\star, x\rangle)\langle w - w^\star, x\rangle].$$

To lower bound the term inside the expectation, consider a particular draw of $x$ and assume $\langle w - w^\star, x\rangle \ge 0$; this is admissible because $h$, like $\rho'$, is odd. Then we have

$$h(\langle w - w^\star, x\rangle)\langle w - w^\star, x\rangle = \frac{h(\langle w - w^\star, x\rangle)}{\langle w - w^\star, x\rangle}\langle w - w^\star, x\rangle^2 \ge c_\rho\langle w - w^\star, x\rangle^2,$$

where the last line follows because $h(0) = 0$ and $h'(0) > c_\rho$. Since this holds pointwise, we simply take the expectation to show that

$$\langle \nabla L_{\mathcal{D}}(w), w - w^\star\rangle \ge c_\rho \mathbb{E}_x\langle w - w^\star, x\rangle^2, \tag{26}$$

and consequently the excess risk is bounded by

$$L_{\mathcal{D}}(w) - L_{\mathcal{D}}(w^\star) \le \frac{C_\rho}{2c_\rho}\langle \nabla L_{\mathcal{D}}(w), w - w^\star\rangle. \tag{27}$$

**Proving the GD conditions.** We now use (27) to establish the GD condition variants.

- $\left(1, \frac{BC_\rho}{c_\rho}\right)$-GD:

  Use Hölder's inequality to obtain the upper bound,

$$\langle \nabla L_{\mathcal{D}}(w), w - w^\star\rangle \le 2B\|\nabla L_{\mathcal{D}}(w)\|.$$

- $\left(2, \frac{C_\rho}{2c_\rho^2 \lambda_{\min}(\Sigma)}\right)$-GD:

  Begin with

  $$L_{\mathcal{D}}(w) - L_{\mathcal{D}}(w^\star) \le \frac{C_\rho}{2c_\rho} \langle \nabla L_{\mathcal{D}}(w), w - w^\star \rangle.$$

  Using the same reasoning as in Lemma 5, this is upper bounded by

  $$\le \frac{C_\rho}{2c_\rho} \|\nabla L_{\mathcal{D}}(w)\|_2 \cdot \|P_{\mathcal{X}}(w - w^\star)\|_2, \tag{28}$$

  where $P_{\mathcal{X}}$ denotes the orthogonal projection onto $\operatorname{span}(\Sigma)$.

  Recalling (26), it also holds that

  $$\begin{aligned}
  \langle P_{\mathcal{X}} \nabla L_{\mathcal{D}}(w), P_{\mathcal{X}}(w - w^\star) \rangle = \langle \nabla L_{\mathcal{D}}(w), w - w^\star \rangle &\ge c_\rho \, \mathbb{E}_x \langle w - w^\star, x \rangle^2 \\
  &= c_\rho \langle w - w^\star, \mathbb{E}_x [xx^T](w - w^\star) \rangle \\
  &= c_\rho \langle w - w^\star, \Sigma(w - w^\star) \rangle \tag{29} \\
  &\ge c_\rho \lambda_{\min}(\Sigma) \|P_{\mathcal{X}}(w - w^\star)\|_2^2.
  \end{aligned}$$

  Rearranging and applying Cauchy-Schwarz, we have

  $$\|P_{\mathcal{X}}(w - w^\star)\|_2 \le \frac{1}{c_\rho \lambda_{\min}(\Sigma)} \cdot \|\nabla \mathcal{L}_{\mathcal{D}}(w)\|_2.$$

  Combining this inequality with (28), we have

  $$L_{\mathcal{D}}(w) - L_{\mathcal{D}}(w^\star) \le \frac{C_\rho}{2c_\rho^2 \lambda_{\min}(\Sigma)} \cdot \|\nabla \mathcal{L}_{\mathcal{D}}(w)\|_2^2.$$

- $\left(2, \frac{2C_\rho s}{c_\rho^2 \psi_{\min}(\Sigma)}\right)$-GD:

  Using the inequality (29) from the $\ell_2/\ell_2$ GD condition proof above

  $$\langle \nabla L_{\mathcal{D}}(w), w - w^\star \rangle \ge c_\rho \langle w - w^\star, \Sigma(w - w^\star) \rangle.$$

  By the assumption that $\|w\|_1 \le \|w^\star\|_1$, we apply Lemma 6 to conclude that 1) $w - w^\star \in \mathcal{C}(S(w^\star), 1)$ and 2) $\|w - w^\star\|_1 \le 2\sqrt{s}\|w - w^\star\|_2$, and so

  $$\langle w - w^\star, \Sigma(w - w^\star) \rangle \ge \psi_{\min}(\Sigma) \|w - w^\star\|_2^2.$$

  Rearranging, and applying the $\|w - w^\star\|_1 \le 2\sqrt{s}\|w - w^\star\|_2$ inequality:

  $$\begin{aligned}
  \|w - w^\star\|_2 &\le \frac{1}{c_\rho \psi_{\min}(\Sigma)} \frac{\langle \nabla L_{\mathcal{D}}(w), w - w^\star \rangle}{\|w - w^\star\|_2} \\
  &\le \frac{1}{c_\rho \psi_{\min}(\Sigma)} \frac{\|\nabla L_{\mathcal{D}}(w)\|_\infty \|w - w^\star\|_1}{\|w - w^\star\|_2} \\
  &\le \frac{2\sqrt{s}}{c_\rho \psi_{\min}(\Sigma)} \|\nabla L_{\mathcal{D}}(w)\|_\infty.
  \end{aligned}$$

  Finally, from (27) we have

  $$\begin{aligned}
  L_{\mathcal{D}}(w) - L_{\mathcal{D}}(w^\star) &\le \frac{C_\rho}{2c_\rho} \langle \nabla L_{\mathcal{D}}(w), w - w^\star \rangle \\
  &\le \frac{C_\rho}{2c_\rho} \|\nabla L_{\mathcal{D}}(w)\|_\infty \|w - w^\star\|_1 \\
  &\le \frac{C_\rho \sqrt{s}}{c_\rho} \|\nabla L_{\mathcal{D}}(w)\|_\infty \|w - w^\star\|_2.
  \end{aligned}$$

  Combining the two inequalities gives the final result.

$\square$

**Lemma 9.** Let the norm $\|\cdot\|$ satisfy the smoothness property (see Theorem 7) with constant $\beta$. Then the gradient for robust regression satisfies the following normed Rademacher complexity bound:

$$\mathbb{E}_\epsilon \sup_{w \in \mathcal{W}} \left\| \sum_{t=1}^n \epsilon_t \nabla \ell(w\,;x_t,y_t) \right\| \leq O\Big(BR^2 C_\rho \sqrt{\beta n}\Big). \tag{30}$$

**Proof of Lemma 9.** Let $G_t(s) = \rho(s - y_t)$ and $F_t(w) = \langle w, x_t \rangle$, so that $\ell(w\,;x_t,y_t) = G_t(F_t(w))$. Then $G_t'(s) = \rho'(s - y_t)$ and $\nabla F_t(w) = x_t$, so our assumptions imply that that $|G_t'(s)| \leq C_\rho$ and $\|\nabla F_t(w)\| \leq R$. We apply Theorem 1 to conclude

$$\mathbb{E}_\epsilon \sup_{w \in \mathcal{W}} \left\| \sum_{t=1}^n \epsilon_t \nabla \ell(w\,;x_t,y_t) \right\| \leq 2R\,\mathbb{E}_\epsilon \sup_{w \in \mathcal{W}} \sum_{t=1}^n \epsilon_t G_t'(\langle w, x_t \rangle) + 2C_\rho \, \mathbb{E}_\epsilon \left\| \sum_{t=1}^n \epsilon_t x_t \right\|.$$

For the first term on the left-hand side, we have that for any $s$, $|G_t''(s)| = 2|\rho''(s - y_t)| \leq 2C_\rho$, so $G_t'$ is $2C_\sigma$-Lipschitz. Then the by scalar contraction for Rademacher complexity (Lemma 1),

$$\mathbb{E}_\epsilon \sup_{w \in \mathcal{W}} \sum_{t=1}^n \epsilon_t G_t'(\langle w, x_t \rangle) \leq 2C_\rho \, \mathbb{E}_\epsilon \sup_{w \in \mathcal{W}} \sum_{t=1}^n \epsilon_t \langle w, x_t \rangle = 2C_\rho B \, \mathbb{E}_\epsilon \left\| \sum_{t=1}^n \epsilon_t x_t \right\|.$$

Finally, the smoothness assumption on the norm (via Theorem 7) implies

$$\mathbb{E}_\epsilon \left\| \sum_{t=1}^n \epsilon_t x_t \right\| \leq \sqrt{2\beta R^2 n}.$$

$\square$

**Proof of Proposition 3.** Observe that Assumption 1 and Assumption 2 respectively imply that $\|\nabla L_{\mathcal{D}}(w^\star)\| = 0$ for the GLM and RR settings. Begin by invoking Theorem 3. It is immediate that any algorithm that guarantees $\mathbb{E}\|\nabla \widehat{L}_n(\widehat{w}^{\mathrm{alg}})\| \leq 1/\sqrt{n}$ will obtain the claimed sample complexity bound (the high-probability statement Theorem 3 immediately yields an in-expectation statement due to boundedness), so all we must do is verify that such a point exists. Proposition 2 along with Lemma 7 and Lemma 9 respectively indeed imply that $\|\nabla \widehat{L}_n(w^\star)\|_2 \leq C/\sqrt{n}$ for both settings.

For completeness, we show below that both models indeed have Lipschitz gradients, and so standard smooth optimizers can be applied to the empirical loss.

*Generalized Linear Model.* Observe that for any $(x, y)$ pair we have

$$\|\nabla \ell(w\,;x,y) - \nabla \ell(w'\,;x,y)\|_2 = 2\|x\|_2 |(\sigma(\langle w, x \rangle) - y)\sigma'(\langle w, x \rangle) - (\sigma(\langle w', x \rangle) - y)\sigma'(\langle w', x \rangle)|.$$

Letting $f(s) = (\sigma(s) - y)\sigma'(s)$, we see that the assumption on the loss guarantees $|f'(s)| \leq 3C_\sigma^2$, so we have

$$\|\nabla \ell(w\,;x,y) - \nabla \ell(w'\,;x,y)\|_2 \leq 6C_\sigma^2 R|\langle w - w', x \rangle| \leq \; \leq 6C_\sigma^2 R^2 \|w - w'\|_2,$$

so smoothness is established.

*Robust Regression.* Following a similar calculation to the GLM case, we have

$$\begin{aligned} \|\nabla \ell(w\,;x,y) - \nabla \ell(w'\,;x,y)\|_2 &= \|x\|_2 |\rho'(\langle w, x \rangle - y) - \rho'(\langle w', x \rangle - y)| \\ &\leq C_\rho \|x\|_2 |\langle w - w^\star, x \rangle| \\ &\leq C_\rho \|x\|_2^2 \|w - w^\star\|_2 \\ &\leq C_\rho R^2 \|w - w^\star\|_2. \end{aligned}$$

Now let $f(s) = (\sigma(s) - y)\sigma'(s)$, and observe that $|f'(s)| \leq 3C_\sigma^2$, so we have

$$\|\nabla \ell(w\,;x,y) - \nabla \ell(w'\,;x,y)\|_2 \leq 6C_\sigma^2 R|\langle w - w', x \rangle| \leq \; \leq 6C_\sigma^2 R^2 \|w - w'\|_2.$$

$\square$

## C.3 Further Discussion

**Detailed comparison with [31]**   We now sketch in more detail the relation between the rates of Theorem 3 and Theorem 4 and those of [31]. We focus on the fast rate regime, and on the case $R = \sqrt{d}$ (e.g., when $x \sim \mathcal{N}(0, I_{d \times d})$).

- *Uniform convergence.* Their uniform convergence bounds scale as $O(\tau \sqrt{d/n})$, where $\tau$ is the subgaussian parameter for the data $x$, whereas our uniform convergence bounds scale as $O(R^2 \sqrt{1/n})$. When $R = \sqrt{d}$ both bounds scale as $O(d\sqrt{1/n})$, but our bounds do not depend on $d$ when $R$ is constant, whereas their bound always pays $\sqrt{d}$.

- *Parameter convergence.* The final result of [31] is a parameter convergence bound of the form $\|\widehat{w}^{\mathrm{alg}} - w^\star\|_2 \le O\left(\frac{\tau}{\gamma \tau^2}\sqrt{\frac{d}{n}}\right)$ (see Theorem 4/6; Eqs. (106) and (96)). Our main result for the "low-dimensional" setup in Theorem 3 and Theorem 4 is an excess risk bound of the form $L_{\mathcal{D}}(\widehat{w}^{\mathrm{alg}}) - L_{\mathcal{D}}(w^\star) \le O\left(\frac{R^4}{\lambda_{\min}(\Sigma)n}\right)$ which implies a parameter convergence bound of $\|\widehat{w}^{\mathrm{alg}} - w^\star\|_2 \le \frac{R^2}{\lambda_{\min}(\Sigma)\sqrt{n}}$ (using similar reasoning as in the proof of Lemma 5 and Lemma 8). With $\tau = R = \sqrt{d}$ and Assumptions 6 and 9 in [31], we have $\lambda_{\min}(\Sigma) = \underline{\gamma}\tau^2$, and so again both the bounds resolve to $O\left(\frac{d}{\lambda_{\min}(\Sigma)\sqrt{n}}\right)$.

**Analysis of regularized stationary point finding for high-dimensional setting**   Here we show that any algorithm that finds a stationary point of the regularized empirical loss generically succeeds obtains optimal sample complexity in the high-dimensional/norm-based setting. We focus on the generalized linear model in the Euclidean setting.

Let $r(w) = \frac{\lambda}{2}\|w\|_2^2$. Define $L_{\mathcal{D}}^\lambda(w) = L_{\mathcal{D}}(w) + r(w)$ and $\widehat{L}_n^\lambda(w) = \widehat{L}_n(w) + r(w)$. We consider any algorithm that returns a point $\widehat{w}$ with $\nabla \widehat{L}_n^\lambda(\widehat{w}) = 0$, i.e. any stationary point of the regularized empirical risk.

**Theorem 9.** *Consider the generalized linear model setting. Let $\widehat{w}$ be any point with $\nabla \widehat{L}_n^\lambda(\widehat{w}) = 0$. Suppose that $\|w^\star\|_2 = 1$ and $C_\sigma, R > 1$. Then there is some absolute constant $c > 0$ such that for any fixed $\delta > 0$, if the regularization parameter $\lambda$ satisfies*

$$\lambda > c \cdot \sqrt{\frac{R^4 C_\sigma^6}{c_\sigma^2} \cdot \frac{\log(\log(C_\sigma R n)/\delta)}{n}},$$

*then with probability at least $1 - \delta$,*

$$L_{\mathcal{D}}(\widehat{w}) - L_{\mathcal{D}}(w^\star) \le O\left(\frac{R^2 C_\sigma^4}{c_\sigma^2} \cdot \sqrt{\frac{\log(\log(C_\sigma R n)/\delta)}{n}}\right).$$

Theorem 9 easily extends to the robust regression setting by replacing invocations of Lemma 7 with Lemma 9 and use of (18) with (27).

**Proof of Theorem 9.**   Recall that $w^\star$ minimizes the *unregularized* population risk, and that $\|w^\star\|_2 = 1$. The technical challenge is to apply Lemma 7 even though we lack a good a-priori upper bound on the norm of $\widehat{w}$. We proceed by splitting the analysis into two cases. The idea is that if $\|\widehat{w}\|_2 \le \|w^\star\|_2$ we can apply Lemma 7 directly with no additional difficulty. On other hand, when $\|\widehat{w}\|_2 \ge \|w^\star\|_2$ the regularized population risk satisfies the $(2, O(1/\lambda))$-GD inequality, which is enough to show that excess risk is small even though $\|\widehat{w}\|_2$ could be larger than $\|w^\star\|_2$.

**Case 1:** $\|\widehat{w}\|_2 \ge \|w^\star\|_2$.

Let $\widetilde{W} = \{w \in \mathbb{R}^d \mid \|w\|_2 \ge \|w^\star\|_2\}$, so that $\widehat{w} \in \widetilde{W}$. Observe that since $r(w)$ is $\lambda$-strongly convex it satisfies $r(w) - r(w^\star) \le \langle \nabla r(w), w - w^\star \rangle - \frac{\lambda}{2}\|w - w^\star\|_2^2$ for all $w$. Moreover, if $w \in \widetilde{W}$, we have

$$\langle \nabla r(w), w - w^\star \rangle \ge r(w) - r(w^\star) + \frac{\lambda}{2}\|w - w^\star\|_2^2 \ge 0.$$

Using (18) and the definition of $w^\star$, along with the strong convexity of $r$, we get

$$L_{\mathcal{D}}^{\lambda}(w) - L_{\mathcal{D}}^{\lambda}(w^\star) \leq \frac{C_\sigma}{2c_\sigma} \langle \nabla L_{\mathcal{D}}(w), w - w^\star \rangle + \langle \nabla r(w), w - w^\star \rangle - \frac{\lambda}{2} \|w - w^\star\|_2^2.$$

Since $\langle \nabla L_{\mathcal{D}}(w), w - w^\star \rangle \geq 0$, this is upper bounded by

$$L_{\mathcal{D}}^{\lambda}(w) - L_{\mathcal{D}}^{\lambda}(w^\star) \leq \frac{C_\sigma}{c_\sigma} \langle \nabla L_{\mathcal{D}}(w), w - w^\star \rangle + \langle \nabla r(w), w - w^\star \rangle - \frac{\lambda}{2} \|w - w^\star\|_2^2.$$

Using the non-negativity of $\langle \nabla r(w), w - w^\star \rangle$ over $\widetilde{W}$, and that $C_\sigma / c_\sigma > 1$, this implies

$$L_{\mathcal{D}}^{\lambda}(w) - L_{\mathcal{D}}^{\lambda}(w^\star) \leq \frac{C_\sigma}{c_\sigma} \langle \nabla L_{\mathcal{D}}^{\lambda}(w), w - w^\star \rangle - \frac{\lambda}{2} \|w - w^\star\|_2^2 \quad \forall w \in \widetilde{W}.$$

Applying Cauchy-Schwarz:

$$\leq \frac{C_\sigma}{c_\sigma} \left\| \nabla L_{\mathcal{D}}^{\lambda}(w) \right\|_2 \|w - w^\star\|_2 - \frac{\lambda}{2} \|w - w^\star\|_2^2 \quad \forall w \in \widetilde{W}.$$

Using the AM-GM inequality:

$$\leq \frac{C_\sigma^2}{c_\sigma^2 \lambda} \left\| \nabla L_{\mathcal{D}}^{\lambda}(w) \right\|_2^2 \quad \forall w \in \widetilde{W}.$$

Using that $\widehat{w} \in \widetilde{W}$, and that $\nabla \widehat{L}_n^{\lambda}(\widehat{w}) = 0$, we have

$$L_{\mathcal{D}}^{\lambda}(\widehat{w}) - L_{\mathcal{D}}^{\lambda}(w^\star) \leq \frac{C_\sigma^2}{c_\sigma^2 \lambda} \left\| \nabla L_{\mathcal{D}}^{\lambda}(\widehat{w}) - \nabla \widehat{L}_n^{\lambda}(\widehat{w}) \right\|_2^2. \tag{31}$$

Observe that since $\widehat{w}$ is a stationary point of the empirical risk, $\nabla \widehat{L}_n(\widehat{w}) = -\lambda \widehat{w}$, and so $\|\widehat{w}\|_2 \leq \frac{1}{\lambda} \left\| \nabla \widehat{L}_n(\widehat{w}) \right\|_2 \leq \frac{2 C_\sigma R}{\lambda}$ with probability 1. Thus, if we apply Lemma 10 with $B_{\max} = \frac{2 C_\sigma R}{\lambda}$, we get that with probability at least $1 - \delta$,

$$\left\| \nabla L_{\mathcal{D}}^{\lambda}(\widehat{w}) - \nabla \widehat{L}_n^{\lambda}(\widehat{w}) \right\|_2 \leq O\left( \|\widehat{w}\|_2 R^2 C_\sigma^2 \sqrt{\frac{1}{n}} + C_\sigma R \sqrt{\frac{\log(\log(C_\sigma R/\lambda)/\delta)}{n}} \right),$$

where we have used additionally that the regularization term does not depend on data. Combining this bound with (31), and using that $\widehat{w} \in \widetilde{W}$ and the elementary inequality $(a+b)^2 \leq 2(a^2 + b^2)$, we see that there exist constants $c, c' > 0$ such that

$$L_{\mathcal{D}}^{\lambda}(\widehat{w}) - L_{\mathcal{D}}^{\lambda}(w^\star) \leq c \cdot \|\widehat{w}\|_2^2 \cdot \frac{R^4 C_\sigma^6}{\lambda c_\sigma^2} \cdot \frac{1}{n} + c' \cdot \frac{R^2 C_\sigma^4}{\lambda c_\sigma^2} \cdot \frac{\log(\log(C_\sigma R/\lambda)/\delta)}{n}.$$

Expanding the definition of the regularized excess risk, this is equivalent to

$$L_{\mathcal{D}}(\widehat{w}) - L_{\mathcal{D}}(w^\star) \leq \lambda + \|\widehat{w}\|_2^2 \cdot \left( c \cdot \frac{R^4 C_\sigma^6}{\lambda c_\sigma^2} \cdot \frac{1}{n} - \lambda \right) + c' \cdot \frac{R^2 C_\sigma^4}{\lambda c_\sigma^2} \cdot \frac{\log(\log(C_\sigma R/\lambda)/\delta)}{n}.$$

Observe that if $\lambda > \sqrt{c \cdot \frac{R^4 C_\sigma^6}{c_\sigma^2} \cdot \frac{1}{n}}$ the middle term in this expression is at most zero. We choose

$$\lambda > \sqrt{c \cdot \frac{R^4 C_\sigma^6}{c_\sigma^2} \cdot \frac{\log(\log(C_\sigma R n)/\delta)}{n}}.$$

Substituting choice this into the expression above leads to a final bound of

$$L_{\mathcal{D}}(\widehat{w}) - L_{\mathcal{D}}(w^\star) \leq O\left( \frac{R^2 C_\sigma^3}{c_\sigma} \cdot \sqrt{\frac{\log(\log(C_\sigma R n)/\delta)}{n}} \right).$$

**Case 2:** $\|\widehat{w}\|_2 \le \|w^\star\|_2$.

Recall that $\nabla\widehat{L}_n^\lambda(\widehat{w}) = 0$. This implies $\nabla\widehat{L}_n(\widehat{w}) = -\lambda\widehat{w}$, and so $\left\|\nabla\widehat{L}_n(\widehat{w})\right\|_2 \le \lambda\|\widehat{w}\|_2 \le \lambda$. Using (18) we have

$$L_{\mathcal{D}}(\widehat{w}) - L_{\mathcal{D}}(w^\star) \le \frac{C_\sigma}{2c_\sigma}\langle\nabla L_{\mathcal{D}}(\widehat{w}), \widehat{w} - w^\star\rangle \le \frac{C_\sigma}{c_\sigma}\|\nabla L_{\mathcal{D}}(\widehat{w})\|_2.$$

Using the bound on the empirical gradient above, we get

$$\|\nabla L_{\mathcal{D}}(\widehat{w})\|_2 \le \lambda + \left\|\nabla L_{\mathcal{D}}(\widehat{w}) - \nabla\widehat{L}_n(\widehat{w})\right\|_2.$$

Using (12), (13), and Lemma 7, applied with $B = 1$, we have that with probability at least $1 - \delta$,

$$\left\|\nabla L_{\mathcal{D}}(\widehat{w}) - \nabla\widehat{L}_n(\widehat{w})\right\|_2 \le O\left(R^2 C_\sigma^2 \sqrt{\frac{\log(1/\delta)}{n}}\right),$$

and so

$$L_{\mathcal{D}}(\widehat{w}) - L_{\mathcal{D}}(w^\star) \le O\left(\lambda\frac{C_\sigma}{c_\sigma} + \frac{R^2 C_\sigma^3}{c_\sigma}\sqrt{\frac{\log(1/\delta)}{n}}\right).$$

Substituting in the choice for $\lambda$:

$$\le O\left(\frac{R^2 C_\sigma^4}{c_\sigma^2} \cdot \sqrt{\frac{\log(\log(C_\sigma Rn)/\delta)}{n}}\right).$$

$\square$

**Lemma 10.** Let $L_{\mathcal{D}}$ and $\widehat{L}_n$ be the population and empirical risk for the generalized linear model setting. Let a parameter $B_{\max} \ge 1$ be given. Then with probability at least $1 - \delta$, for all $w \in \mathbb{R}^d$ with $1 \le \|w\|_2 \le B_{\max}$,

$$\left\|\nabla L_{\mathcal{D}}(w) - \nabla\widehat{L}_n(w)\right\|_2 \le O\left(\|w\|_2 R^2 C_\sigma^2\sqrt{\frac{1}{n}} + C_\sigma R\sqrt{\frac{\log(\log(B_{\max})/\delta)}{n}}\right),$$

where all constants are as in Assumption 1.

**Proof.** (12), (13), and Lemma 7 imply that for any fixed $B$, with probability at least $1 - \delta$,

$$\sup_{w:\|w\|_2\le B}\left\|\nabla L_{\mathcal{D}}(w) - \nabla\widehat{L}_n(w)\right\| \le O\left(BR^2 C_\sigma^2\sqrt{\frac{1}{n}} + C_\sigma R\sqrt{\frac{\log(\frac{1}{\delta})}{n}}\right).$$

Define $B_i = e^{i-1}$ for $1 \le i \le \lceil\log(B_{\max})\rceil + 1$. The via a union bound, we have that for all $i$ simultaneously,

$$\sup_{w:\|w\|_2\le B_i}\left\|\nabla L_{\mathcal{D}}(w) - \nabla\widehat{L}_n(w)\right\| \le O\left(B_i R^2 C_\sigma^2\sqrt{\frac{1}{n}} + C_\sigma R\sqrt{\frac{\log(\log(B_{\max})/\delta)}{n}}\right).$$

In particular, for any fixed $w$ with $1 \le \|w\|_2 \le B_{\max}$, if we take $i$ to be the smallest index for which $\|w\|_2 \le B_i$, the expression above implies

$$\left\|\nabla L_{\mathcal{D}}(w) - \nabla\widehat{L}_n(w)\right\|_2 \le O\left(\|w\|_2 R^2 C_\sigma^2\sqrt{\frac{1}{n}} + C_\sigma R\sqrt{\frac{\log(\log(B_{\max})/\delta)}{n}}\right),$$

since $B_i \le e\|w\|_2$. $\square$

**Analysis of mirror descent for high-dimensional setting.** Here we show that mirror descent obtains optimal excess risk for the norm-based/high-dimensional regime in Theorem 3 and Theorem 4.

Our approach is to run mirror descent with $\Psi^\star$ as the regularizer. Observe that $\Psi^\star$ is $\frac{1}{\beta}$-strongly convex with respect to the dual norm $\|\cdot\|_\star$, and that we have $\|\nabla\ell(w\,;x,y)\| \leq 2C_\sigma R$ for the GLM setting and $\|\nabla\ell(w\,;x,y)\| \leq C_\rho R$ for the RR setting.

Focusing on the GLM, if we take a single pass over the entire dataset $\{(x_t, y_t)\}_{t=1}^n$ in order, the standard analysis for mirror descent starting at $w_1 = 0$ with optimal learning rate tuning [15] guarantees that the following inequality holds deterministically:

$$\frac{1}{n}\sum_{t=1}^n \langle \nabla\ell(w_t\,;x_t,y_t), w_t - w^\star \rangle \leq O\left( RBC_\sigma \sqrt{\frac{\beta}{n}} \right).$$

Since each point is visited a single time, this leads to the following guarantee on the population loss in expectation

$$\mathbb{E}\left[ \frac{1}{n}\sum_{t=1}^n \langle \nabla L_\mathcal{D}(w_t), w_t - w^\star \rangle \right] \leq O\left( RBC_\sigma \sqrt{\frac{\beta}{n}} \right).$$

Consequently, if we define $\widehat{w}$ to be the result of choosing a single time $t \in [n]$ uniformly at random and returning $w_t$, this implies that

$$\mathbb{E}[\langle \nabla L_\mathcal{D}(\widehat{w}), \widehat{w} - w^\star \rangle] \leq O\left( RBC_\sigma \sqrt{\frac{\beta}{n}} \right).$$

Combining this inequality with (18), we have

$$\mathbb{E}[L_\mathcal{D}(\widehat{w}) - L_\mathcal{D}(w^\star)] \leq O\left( RB\frac{C_\sigma^2}{c_\sigma} \sqrt{\frac{\beta}{n}} \right).$$

Likewise, combining the mirror descent upper bound with (27) leads to a rate of $O\left( RB\frac{C_\rho^2}{c_\rho}\sqrt{\frac{\beta}{n}} \right)$ for robust regression. Thus, when all parameters involved are constant, it suffices to take $n = \frac{1}{\varepsilon^2}$ to obtain $O(\varepsilon)$ excess risk in both settings.

## D  Proofs from Section 4

**Proof of Theorem 5.** Let $B \in \mathbb{R}^{d\times d}$ be a matrix for which the $i$th row $B_i$ is given by $B_i = \frac{1}{\sqrt{d}}(\mathbf{1} - e_i)$.

We first focus on the more technical case where $n \geq d$.

Let $n = N \cdot d$ for some odd $N \in \mathbb{N}$. We partition time into $d$ consecutive segments: $S_1 = \{1, \ldots, N\}$, $S_2 = \{N+1, \ldots, 2N\}$ and on. The sequence of instances $x_{1:n}$ we will use will be to set $x_t = B_i$ for $t \in S_i$. Note that $\|B_i\|_2 \leq 1$, so this choice indeed satisfies the boundedness constraint.

For simplicity, assume that $y_t = -1$ for all $t \in [n]$. Then it holds that

$$\mathbb{E}_\epsilon \sup_{w\in\mathcal{W}}\left\| \sum_{t=1}^n \epsilon_t \nabla\ell(w\,;x_t,y_t) \right\|_2 = \mathbb{E}_\epsilon \sup_{w\in\mathcal{W}}\left\| \sum_{t=1}^n \epsilon_t \mathbb{1}\{\langle w, x_t\rangle \geq 0\}x_t \right\|_2$$

$$= \mathbb{E}_\epsilon \sup_{w\in\mathcal{W}}\left\| \sum_{i=1}^d \mathbb{1}\{\langle w, B_i\rangle \geq 0\} \sum_{t\in S_i} \epsilon_t x_t \right\|_2$$

We introduce the notation $\varphi_i = \sum_{t\in S_i}\epsilon_t$.

$$= \mathbb{E}_\varphi \sup_{w\in\mathcal{W}}\left\| \sum_{i=1}^d \mathbb{1}\{\langle w, B_i\rangle \geq 0\}\varphi_i B_i \right\|_2$$

$$= \mathbb{E}_\varphi \sup_{w\in\mathcal{W}}\left\| \sum_{i=1}^d \mathbb{1}\{\langle w, B_i\rangle \geq 0\}\varphi_i \frac{1}{\sqrt{d}}(\mathbf{1} - e_i) \right\|_2$$

Using triangle inequality:

$$\geq \mathbb{E}_\varphi \sup_{w \in \mathcal{W}} \left\| \sum_{i=1}^d \mathbb{1}\{\langle w, B_i \rangle \geq 0\} \varphi_i \frac{1}{\sqrt{d}} \mathbf{1} \right\|_2 - \frac{1}{\sqrt{d}} \mathbb{E}_\varphi \sum_{i=1}^d |\varphi_i|$$

$$= \mathbb{E}_\varphi \sup_{w \in \mathcal{W}} \left| \sum_{i=1}^d \mathbb{1}\{\langle w, B_i \rangle \geq 0\} \varphi_i \right| - \frac{1}{\sqrt{d}} \mathbb{E}_\varphi \sum_{i=1}^d |\varphi_i|$$

$$\geq \mathbb{E}_\varphi \sup_{w \in \mathcal{W}} \left| \sum_{i=1}^d \mathbb{1}\{\langle w, B_i \rangle \geq 0\} \varphi_i \right| - O(\sqrt{n}).$$

Now, for a given draw of $\varphi$, we choose $w \in \mathcal{W}$ such that $\mathrm{sgn}(\langle w, B_i \rangle) = \mathrm{sgn}(\varphi_i)$. The key trick here is that $B$ is invertible, so for a given sign pattern, say $\sigma \in \{\pm 1\}^d$, we can set $\widetilde{w} = B^{-1}\sigma$ and then $w = \widetilde{w}/\|\widetilde{w}\|_2$ to achieve this pattern. To see that $B$ is invertible, observe that we can write it as $B = \frac{1}{\sqrt{d}}(\mathbf{1}\mathbf{1}^\top - I)$. The identity matrix can itself be written as $\frac{1}{d}\mathbf{1}\mathbf{1}^\top + A_\perp$, where $\mathbf{1} \notin \mathrm{span}(A_\perp)$, so it can be seen that $B = \frac{1}{\sqrt{d}}\big((1 - \frac{1}{d})\mathbf{1}\mathbf{1}^\top - A_\perp\big)$, and that the $\mathbf{1}\mathbf{1}^\top$ component is preserved by this addition.

We have now arrived at a lower bound of $\mathbb{E}_\varphi \big| \sum_{i=1}^d \mathbb{1}\{\mathrm{sgn}(\varphi_i) \geq 0\} \varphi_i \big|$. This value is lower bounded by

$$\mathbb{E}_\varphi \left| \sum_{i=1}^d \mathbb{1}\{\mathrm{sgn}(\varphi_i) \geq 0\} \varphi_i \right|$$

$$= \mathbb{E}_\varphi \sum_{i=1}^d \mathbb{1}\{\mathrm{sgn}(\varphi_i) \geq 0\} |\varphi_i|$$

Now, observe that since $N$ is odd we have $\mathrm{sgn}(\varphi_i) \in \{\pm 1\}$, and so $\mathbb{1}\{\mathrm{sgn}(\varphi_i) \geq 0\} = (1 + \mathrm{sgn}(\varphi_i))/2$. Furthermore, since $\varphi_i$ is symmetric, we may replace $\mathrm{sgn}(\varphi_i)$ with an independent Rademacher random variable $\sigma_i$

$$= \mathbb{E}_\varphi \mathbb{E}_\sigma \frac{1}{2} \sum_{i=1}^d (1 + \sigma_i) |\varphi_i|$$

$$= \mathbb{E}_\varphi \frac{1}{2} \sum_{i=1}^d |\varphi_i|.$$

Lastly, the Khintchine inequality implies that $\mathbb{E}_{\varphi_i} |\varphi_i| \geq \sqrt{N/2}$, so the final lower bound is $\Omega(d\sqrt{N}) = \Omega(\sqrt{dn})$.

In the case where $d \geq n$, the argument above easily yields that $\mathbb{E}_\epsilon \sup_{w \in \mathcal{W}} \| \sum_{t=1}^n \epsilon_t \nabla \ell(w; x_t, y_t) \|_2 = \Omega(n)$.

$\square$

### D.1 Proof of Theorem 6

Before proceeding to the proof, let us introduce some auxiliary definitions and results. The following functions will be used throughout the proof. They are related by Lemma 11.

$$\xi_{\mathcal{D}}(w, \gamma) = \mathbb{E}_{x \sim \mathcal{D}} \, \mathbb{1}\left\{ \frac{|\langle w, x \rangle|}{\|w\|\|x\|} \leq \gamma \right\},$$

$$\widehat{\xi}_n(w, \gamma) = \frac{1}{n} \sum_{t=1}^n \mathbb{1}\left\{ \frac{|\langle w, x_t \rangle|}{\|w\|\|x_t\|} \leq \gamma \right\}.$$

**Lemma 11.** With probability at least $1 - \delta$, simultaneously for all $w \in \mathcal{W}$ and all $\gamma > 0$,

$$\xi_{\mathcal{D}}(w, \gamma) \leq \widehat{\xi}_n(w, 2\gamma) + \frac{4}{\gamma\sqrt{n}} + \sqrt{\frac{2\log(\log_2(4/\gamma)/\delta)}{n}},$$

$$\widehat{\xi}_n(w, \gamma) \leq \xi_{\mathcal{D}}(w, 2\gamma) + \frac{4}{\gamma\sqrt{n}} + \sqrt{\frac{2\log(\log_2(4/\gamma)/\delta)}{n}}.$$

**Proof sketch for Lemma 11.** We only sketch the proof here as it follows standard analysis (see Theorem 5 of [20]). The key technique is to introduce a Lipschitz function $\zeta_\gamma(t)$:

$$\zeta_\gamma(t) = \begin{cases} 1 & |t| \le \gamma \\ 2 - |t|/\gamma & \gamma < |t| < 2\gamma \\ 0 & |t| \ge 2\gamma \end{cases}.$$

Observe that $\zeta_\gamma$ satisfies $\mathbb{1}\{|t| > \gamma\} \le \zeta_\gamma(t) \le \mathbb{1}\{|t| > 2\gamma\}$ for all $t$. This sandwiching allows us to bound $\sup_{w \in \mathcal{W}}\{\xi_\mathcal{D}(w, \gamma) - \widehat{\xi}_n(w, 2\gamma)\}$ (and $\sup_{w \in \mathcal{W}}\{\widehat{\xi}_n(w, \gamma) - \xi_\mathcal{D}(w, 2\gamma)\}$ ) by instead bounding the difference between the empirical and population averages of the surrogate $\zeta_\gamma$. This is achieved easily using the Lipschitz contraction lemma for Rademacher complexity, and by noting that the Rademacher complexity of the class $\{x \mapsto \langle w, x\rangle \mid \|w\|_2 \le 1\}$ is at most $\sqrt{n}$ whenever data satisfies $\|x_t\|_2 \le 1$ for all $t$. Finally, a union bound over values of $\gamma$ in the range $[0, 1]$ yields the statement. $\square$

**Proof of Theorem 6.** Let the margin function $\phi$ and $\delta > 0$ be fixed. Define functions $\psi(\cdot)$, $\phi_1(\cdot)$, and $\phi_2(\cdot)$ as follows:

$$\psi(\gamma) = \frac{4}{\gamma\sqrt{n}} + \sqrt{\frac{2\log(\log_2(4/\gamma)/\delta)}{n}}$$
$$\phi_1(\gamma) = \phi(2\gamma) + \psi(\gamma)$$
$$\phi_2(\gamma) = \phi(4\gamma) + 2\psi(2\gamma).$$

Now, conditioning on the events of Lemma 11, we have that with probability at least $1 - \delta$,

$$\mathcal{W}(\phi, \widehat{\mathcal{D}}_n) \subseteq \mathcal{W}(\phi_1, \mathcal{D}) \subseteq \mathcal{W}(\phi_2, \widehat{\mathcal{D}}_n). \tag{32}$$

Consequently, we have the upper bound

$$\sup_{w \in \mathcal{W}(\phi, \widehat{\mathcal{D}}_n)} \left\| \nabla L_\mathcal{D}(w) - \nabla \widehat{L}_n(w) \right\|_2 \le \sup_{w \in \mathcal{W}(\phi_1, \mathcal{D})} \left\| \nabla L_\mathcal{D}(w) - \nabla \widehat{L}_n(w) \right\|_2$$

$$\le 4\,\mathbb{E}_\epsilon \sup_{w \in \mathcal{W}(\phi_1, \mathcal{D})} \left\| \frac{1}{n}\sum_{t=1}^n \epsilon_t \nabla \ell(w\,; x_t, y_t) \right\| + 4\sqrt{\frac{\log(2/\delta)}{n}}$$

$$\le 4\,\mathbb{E}_\epsilon \sup_{w \in \mathcal{W}(\phi_2, \widehat{\mathcal{D}}_n)} \left\| \frac{1}{n}\sum_{t=1}^n \epsilon_t \nabla \ell(w\,; x_t, y_t) \right\| + 4\sqrt{\frac{\log(2/\delta)}{n}}, \tag{33}$$

where the second inequality holds with probability at least $1 - \delta$ using Lemma 4. They key here is that we are able to apply the standard symmetrization result because we have replaced $\mathcal{W}(\phi, \widehat{\mathcal{D}}_n)$ with a set that does not depend on data. Next, invoking the chain rule (Theorem 1), we split the Rademacher complexity term above as:

$$\mathbb{E}_\epsilon \sup_{w \in \mathcal{W}(\phi_2, \widehat{\mathcal{D}}_n)} \left\| \frac{1}{n}\sum_{t=1}^n \epsilon_t \nabla \ell(w\,; x_t, y_t) \right\| \le 2\,\mathbb{E}_\epsilon \underbrace{\sup_{w \in \mathcal{W}(\phi_2, \widehat{\mathcal{D}}_n)} \frac{1}{n}\sum_{t=1}^n \epsilon_t \mathbb{1}\{y_t\langle w, x_t\rangle \le 0\}}_{(\blacklozenge)} + \frac{2}{n}\,\mathbb{E}_\epsilon \left\| \sum_{t=1}^n \epsilon_t x_t \right\|_2. \tag{34}$$

The second term is controlled by Theorem 7, which gives $\frac{1}{n}\,\mathbb{E}_\epsilon \|\sum_{t=1}^n \epsilon_t x_t\|_2 \le \frac{1}{\sqrt{n}}$. For the first term, we appeal to the fat-shattering dimension and the $\phi_2$-soft-margin assumption.

**Controlling ($\blacklozenge$).** Observe that for any fixed $\tilde{\gamma} > 0$, we can split ($\blacklozenge$) as

$$\mathbb{E}_\epsilon \sup_{w \in \mathcal{W}(\phi_2, \widehat{\mathcal{D}}_n)} \frac{1}{n}\sum_{t=1}^n \epsilon_t \mathbb{1}\{y_t\langle w, x_t\rangle \le 0\} \le \underbrace{\mathbb{E}_\epsilon \sup_{w \in \mathcal{W}(\phi_2, \widehat{\mathcal{D}}_n)} \frac{1}{n}\sum_{t=1}^n \epsilon_t \mathbb{1}\left\{y_t\langle w, x_t\rangle \le 0 \wedge \frac{|\langle w, x_t\rangle|}{\|w\|_2\|x_t\|_2} \ge \tilde{\gamma}\right\}}_{(\star)}$$

$$+ \underbrace{\mathbb{E}_\epsilon \sup_{w \in \mathcal{W}(\phi_2, \widehat{\mathcal{D}}_n)} \frac{1}{n}\sum_{t=1}^n \epsilon_t \mathbb{1}\left\{y_t\langle w, x_t\rangle \le 0 \wedge \frac{|\langle w, x_t\rangle|}{\|w\|_2\|x_t\|_2} < \tilde{\gamma}\right\}}_{(\star\star)}.$$

For ($\star\star$), the definition of $\mathcal{W}(\phi_2, \widehat{\mathcal{D}}_n)$ implies

$$\mathbb{E}_\epsilon \sup_{w \in \mathcal{W}(\phi_2, \widehat{\mathcal{D}}_n)} \frac{1}{n} \sum_{t=1}^n \epsilon_t \mathbb{1}\Big\{ y_t \langle w, x_t \rangle \le 0 \wedge \frac{|\langle w, x_t \rangle|}{\|w\|_2 \|x_t\|_2} < \tilde{\gamma} \Big\} \le \sup_{w \in \mathcal{W}(\phi_2, \widehat{\mathcal{D}}_n)} \frac{1}{n} \sum_{t=1}^n \mathbb{1}\Big\{ \frac{|\langle w, x_t \rangle|}{\|w\|_2 \|x_t\|_2} < \tilde{\gamma} \Big\} \le \phi_2(\tilde{\gamma}).$$
(35)

The quantity ($\star$) can be bounded by writing it as

$$\mathbb{E}_\epsilon \sup_{v \in V} \frac{1}{n} \sum_{t=1}^n \epsilon_t v_t,$$

where $V$ is a boolean concept class defined as $V = \big\{ (v_1(w), \ldots, v_n(w)) \in \{0,1\}^n \mid w \in \mathcal{W}(\phi_2, \widehat{\mathcal{D}}_n) \big\}$, where $v_i(w) := \mathbb{1}\big\{ y_i \frac{\langle w, x_i \rangle}{\|w\|_2 \|x_i\|_2} \le 0 \big\} \cdot \mathbb{1}\big\{ \frac{|\langle w, x_i \rangle|}{\|w\|_2 \|x_i\|_2} \ge \tilde{\gamma} \big\}$. The standard Massart finite class lemma (e.g. [32]) implies

$$\mathbb{E}_\epsilon \sup_{v \in V} \frac{1}{n} \sum_{t=1}^n \epsilon_t v_t \le \sqrt{\frac{2 \log |V|}{n}}.$$

All that remains is to bound the cardinality of $V$. To this end, note that we can bound $|V|$ by first counting the number of realizations of $\Big( \mathbb{1}\big\{ \frac{|\langle w, x_1 \rangle|}{\|w\|_2 \|x_1\|_2} \ge \tilde{\gamma} \big\}, \ldots, \mathbb{1}\big\{ \frac{|\langle w, x_n \rangle|}{\|w\|_2 \|x_n\|_2} \ge \tilde{\gamma} \big\} \Big)$ as we vary $w \in \mathcal{W}(\phi_2, \widehat{\mathcal{D}}_n)$. This is at most $\binom{n}{n\phi_2(\tilde{\gamma})} \le n^{n\phi_2(\tilde{\gamma})}$, since the number of points with margin smaller than $\tilde{\gamma}$ is bounded by $n\phi_2(\tilde{\gamma})$ via (32).

Next, we consider only the points $x_t$ for which $\mathbb{1}\big\{ \frac{|\langle w, x_t \rangle|}{\|w\|_2 \|x_1\|_2} \ge \tilde{\gamma} \big\} = 1$. On these points, on which we are guaranteed to have a margin at least $\tilde{\gamma}$, we count the number of realizations of $\mathbb{1}\big\{ y_t \frac{\langle w, x_t \rangle}{\|w\|_2 \|x_t\|_2} \le 0 \big\}$. This is bounded by $n^{O\left(\frac{1}{\tilde{\gamma}^2}\right)}$ due to the Sauer-Shelah lemma (e.g. [39]). The fat-shattering dimension at margin $\tilde{\gamma}$ coincides with the notion of shattering on these points, and [4] bound the fat-shattering dimension at scale $\tilde{\gamma}$ by $O\left(\frac{1}{\tilde{\gamma}^2}\right)$. Hence, the cardinality of $V$ is bounded by

$$|V| \le n^{n\phi_2(\tilde{\gamma})} n^{O\left(\frac{1}{\tilde{\gamma}^2}\right)}.$$
(36)

**Final bound.** Assembling equations (33), (34), (35), and (36) yields

$$\sup_{w \in \mathcal{W}(\phi, \widehat{\mathcal{D}}_n)} \big\| \nabla L_{\mathcal{D}}(w) - \nabla \widehat{L}_n(w) \big\|_2 \le O\left( \phi_2(\tilde{\gamma}) + \sqrt{\frac{\log |V|}{n}} + \sqrt{\frac{\log(1/\delta)}{n}} \right)$$

$$\le O\left( \phi_2(\tilde{\gamma}) + \sqrt{\left( \phi_2(\tilde{\gamma}) + \frac{1}{\tilde{\gamma}^2 n} \right) \log(n)} + \sqrt{\frac{\log(1/\delta)}{n}} \right)$$

$$\le \tilde{O}\left( \sqrt{\phi_2(\tilde{\gamma})} + \frac{1}{\tilde{\gamma}\sqrt{n}} + \sqrt{\frac{\log(1/\delta)}{n}} \right)$$

$$\le \tilde{O}\left( \sqrt{\phi(4\tilde{\gamma})} + \frac{1}{\tilde{\gamma}} \sqrt{\frac{\log(1/\delta)}{n}} + \frac{1}{\sqrt{\tilde{\gamma}} n^{1/4}} \right).$$

The chain of inequalities above follows by observing that $\phi_2(\tilde{\gamma}) = \phi(4\tilde{\gamma}) + 2\psi(2\tilde{\gamma})$ is bounded and thus $\phi_2(\gamma) \le c\sqrt{\phi_2(\gamma)}$ for some constant $c$ independent of $\tilde{\gamma}$. We get the desired result by optimizing over $\tilde{\gamma}$.

$\square$

# E   Additional Results

**Theorem 10** (Second-order chain rule for Rademacher complexity)**.** *Let two sequences of twice-differentiable functions $G_t : \mathbb{R}^K \to \mathbb{R}$ and $F_t : \mathbb{R}^d \to \mathbb{R}^K$ be given, and let $F_{t,i}(w)$ denote the ith of*

*coordinate of $F_t(w)$. Suppose there are constants $L_{F,1}$, $L_{F,2}$, $L_{G,1}$, $L_{G,2}$ such that for all $1 \le t \le n$, $\|\nabla G_t\|_2 \le L_{G,1}$, $\sqrt{\sum_{i,j}\|(\nabla F_{t,i})(\nabla F_{t,j})^\top\|_\sigma^2} \le L_{F,1}$, $\|\nabla^2 G_t\|_2 \le L_{G,2}$ and $\sqrt{\sum_{k=1}^K \|\nabla^2 F_{t,k}\|_\sigma^2} \le L_{F,2}$. Then,*

$$\frac{1}{2}\,\mathbb{E}_\epsilon \sup_{w\in\mathcal{W}}\left\|\sum_{t=1}^n \epsilon_t \nabla^2(G_t(F_t(w)))\right\|_\sigma \le L_{F,1}\,\mathbb{E}_\epsilon \sup_{w\in\mathcal{W}}\sum_{t=1}^n \langle \epsilon_t, \nabla G_t(F_t(w))\rangle$$
$$+ L_{G,1}\,\mathbb{E}_\epsilon \sup_{w\in\mathcal{W}}\left\|\sum_{t=1}^n \sum_{i=1}^K \epsilon_{t,k}\nabla^2 F_{t,i}(w)\right\|_\sigma$$
$$+ L_{F,2}\,\mathbb{E}_{\tilde\epsilon} \sup_{w\in\mathcal{W}}\sum_{t=1}^n \langle \tilde\epsilon_t, \nabla^2 G_t(F_t(w))\rangle$$
$$+ L_{G,2}\,\mathbb{E}_{\tilde\epsilon} \sup_{w\in\mathcal{W}}\left\|\sum_{t=1}^n \sum_{i=1,j=1}^K \tilde\epsilon_{t,i,j}\nabla F_{t,i}(w)\nabla F_{t,j}(w)^\top\right\|_\sigma,$$

*where for all $i \in [K]$, $\nabla F_{t,i}(w)$ denotes the $i^{th}$ column of the Jacobian matrix $\nabla F_t \in \mathbb{R}^{d\times K}$, $\nabla^2 F_{t,i} \in \mathbb{R}^{d\times d}$ denotes the ith slice of the Hessian operator $\nabla^2 F_t \in \mathbb{R}^{d\times d\times K}$, and $\epsilon \in \{\pm 1\}^{n,k}$ and $\tilde\epsilon \in \{\pm 1\}^{n\times K\times K}$ are matrices of Rademacher random variables.*

As an application of Theorem 10, we give a simple proof of dimension-independent Rademacher bound for the generalized linear model setting.

**Lemma 12.** Assume in addition to Assumption 1 assume that $|\sigma'''(s)| \le C_\sigma$ for all $s \in \mathcal{S}$, and suppose $\|\cdot\|$ is any $\beta$-smooth norm. Then the empirical loss Hessian for the generalized linear model setting enjoys the normed Rademacher complexity bound,

$$\mathbb{E}_\epsilon \sup_{w\in\mathcal{W}}\left\|\sum_{t=1}^n \epsilon_t \nabla^2 \ell(w\,;x_t,y_t)\right\|_\sigma \le O\Big(\big(BR^3 C_\sigma^2 \sqrt{\beta} + C_\sigma^2 R^2 \sqrt{\log(d)}\big)\sqrt{n}\Big). \tag{37}$$

It is easy to see that the same approach leads to a normed Rademacher complexity bound for the Hessian in the robust regression setting as well. We leave the proof as an exercise.

**Lemma 13.** Assume in addition to Assumption 2 that $|\rho'''(s)| \le C_\rho$ for all $s \in \mathcal{S}$, and suppose $\|\cdot\|$ is any $\beta$-smooth norm. Then the empirical loss Hessian for the robust regression setting enjoys the normed Rademacher complexity bound:

$$\mathbb{E}_\epsilon \sup_{w\in\mathcal{W}}\left\|\sum_{t=1}^n \epsilon_t \nabla^2 \ell(w\,;x_t,y_t)\right\|_\sigma \le O\Big(\big(BR^3 C_\rho \sqrt{\beta} + C_\rho R^2 \sqrt{\log(d)}\big)\sqrt{n}\Big). \tag{38}$$

**Proof of Theorem 10.** We start by writing

$$\mathbb{E}_\epsilon \sup_{w\in\mathcal{W}}\left\|\sum_{t=1}^n \epsilon_t \nabla^2(G_t(F_t(w)))\right\|_\sigma = \mathbb{E}_\epsilon \sup_{w\in\mathcal{W}} \sup_{\substack{u\in\mathbb{R}^d \\ \|u\|_2\le 1}}\sum_{t=1}^n \epsilon_t u^\top \nabla^2(G_t(F_t(w)))u. \tag{39}$$

Using the chain rule for differentiation, we have for any $u \in \mathbb{R}^n$

$$u^\top \nabla^2(G_t(F_t(w)))u = \langle \nabla F_t(w), u\rangle^\top \nabla^2 G_t(F_t(w)) \langle \nabla F_t(w), u\rangle + \langle \nabla G_t(F_t(w)), \nabla^2 F_t(w)[u,u]\rangle,$$

where $\nabla G_t(F_t(w))$ and $\nabla^2 G_t(F_t(w))$ denote the gradient and Hessian of $G_t$ at $F_t(w)$, and $\nabla^2 F_t(w)[u,u] \in \mathbb{R}^K$ is a vector for which the $i$th coordinate is the evaluation of the Hessian operator for $F_{t,i}$ at $(u,u)$. Using this identity along with (39), we get

$$\mathbb{E}_\epsilon \sup_{w\in\mathcal{W}}\left\|\sum_{t=1}^n \epsilon_t \nabla^2(G_t(F_t(w)))\right\|_\sigma \le \mathbb{E}_\epsilon \sup_{w\in\mathcal{W}} \sup_{\substack{u\in\mathbb{R}^d \\ \|u\|_2\le 1}}\sum_{t=1}^n \epsilon_t \langle \nabla G_t(F_t(w)), \nabla^2 F_t(w)[u,u]\rangle$$
$$+ \mathbb{E}_\epsilon \sup_{w\in\mathcal{W}} \sup_{\substack{u\in\mathbb{R}^d \\ \|u\|_2\le 1}}\sum_{t=1}^n \epsilon_t \langle \nabla F_t(w), u\rangle^\top \nabla^2 G_t(F_t(w)) \langle \nabla F_t(w), u\rangle.$$

We bound the two terms separately.

1. *First Term:* We introduce a new function that relabels the quantities in the expression. Let $h_1 : \mathbb{R}^{2K} \to \mathbb{R}$ be defined as $h_1(a,b) = \langle a, b \rangle$, let $f_1 : \mathcal{W} \times \mathbb{R}^d \to \mathbb{R}^K$ be given by $f_1(w,u) = \nabla G_t(F_t(w))$ and $f_2 : \mathcal{W} \times \mathbb{R}^d \to \mathbb{R}^K$ be given by $f_2(w,u) = (\nabla^2 F_{t,k}(w)[u,u])_{k \in [K]}$. We apply the block-wise contraction lemma Lemma 3 with one block for $f_1$ and one block for $f_2$ to conclude

$$\frac{1}{2} \mathbb{E}_\epsilon \sup_{w \in \mathcal{W}} \sup_{\substack{u \in \mathbb{R}^d \\ \|u\|_2 \leq 1}} \sum_{t=1}^n \epsilon_t h_1(f_1(w,u), f_2(w,u))$$

$$\leq L_{F,1} \mathbb{E}_\epsilon \sup_{w \in \mathcal{W}} \sup_{\substack{u \in \mathbb{R}^d \\ \|u\|_2 \leq 1}} \sum_{t=1}^n \langle \boldsymbol{\epsilon}_t, f_1(w,u) \rangle + L_{G,1} \mathbb{E}_\epsilon \sup_{w \in \mathcal{W}} \sup_{\substack{u \in \mathbb{R}^d \\ \|u\|_2 \leq 1}} \sum_{t=1}^n \langle \boldsymbol{\epsilon}_t, f_2(w,u) \rangle$$

$$\leq L_{F,1} \mathbb{E}_\epsilon \sup_{w \in \mathcal{W}} \sup_{\substack{u \in \mathbb{R}^d \\ \|u\|_2 \leq 1}} \sum_{t=1}^n \langle \boldsymbol{\epsilon}_t, \nabla G_t(F_t(w)) \rangle + L_{G,1} \mathbb{E}_\epsilon \sup_{w \in \mathcal{W}} \sup_{\substack{u \in \mathbb{R}^d \\ \|u\|_2 \leq 1}} \sum_{t=1}^n \langle \boldsymbol{\epsilon}_t, \nabla^2 F_t(w)[u,u] \rangle$$

$$\leq L_{F,1} \mathbb{E}_\epsilon \sup_{w \in \mathcal{W}} \sum_{t=1}^n \langle \boldsymbol{\epsilon}_t, \nabla G_t(F_t(w)) \rangle + L_{G,1} \mathbb{E}_\epsilon \sup_{w \in \mathcal{W}} \sup_{\substack{u \in \mathbb{R}^d \\ \|u\|_2 \leq 1}} \left( \sum_{t=1}^n \nabla^2 F_t(w) \boldsymbol{\epsilon}_t \right)[u,u]$$

$$= L_{F,1} \mathbb{E}_\epsilon \sup_{w \in \mathcal{W}} \sum_{t=1}^n \langle \boldsymbol{\epsilon}_t, \nabla G_t(F_t(w)) \rangle + L_{G,1} \mathbb{E}_\epsilon \sup_{w \in \mathcal{W}} \left\| \sum_{t=1}^n \nabla^2 F_t(w) \boldsymbol{\epsilon}_t \right\|_\sigma.$$

2. *Second Term:* Let us first simplify as

$$\langle \nabla F_t(w), u \rangle^\top \nabla^2 G_t(F_t(w)) \langle \nabla F_t(w), u \rangle = \sum_{i,j=1}^K \langle \nabla F_t(w), u \rangle_i \nabla^2 G_t(F_t(w))_{i,j} \langle \nabla F_t(w), u \rangle_j$$

$$= \sum_{i,j=1}^K \left( u^\top \nabla F_{t,i}(w) \right) \times \frac{\partial^2 G_t}{\partial z_i \partial z_j} \times \left( \nabla F_{t,j}(w)^\top u \right)$$

$$= \sum_{i,j=1}^K \frac{\partial^2 G_t}{\partial z_i \partial z_j} \left( u^\top \nabla F_{t,i}(w) \nabla F_{t,j}(w)^\top u \right),$$

where $\nabla F_{t,j}(w) := \nabla F_t(w)[:,j] \in \mathbb{R}^d$, and the last equality follows by observing that $\frac{\partial^2 G_t}{\partial z_i \partial z_j}$ is scalar. We thus have

$$\mathbb{E}_\epsilon \sup_{w \in \mathcal{W}} \sup_{\substack{u \in \mathbb{R}^d \\ \|u\|_2 \leq 1}} \sum_{t=1}^n \epsilon_t \langle \nabla F_t(w), u \rangle^\top \nabla^2 G_t(F_t(w)) \langle \nabla F_t(w), u \rangle$$

$$= \mathbb{E}_\epsilon \sup_{w \in \mathcal{W}} \sup_{\substack{u \in \mathbb{R}^d \\ \|u\|_2 \leq 1}} \sum_{t=1}^n \epsilon_t \sum_{i,j=1}^K \frac{\partial^2 G_t}{\partial z_i \partial z_j} \left( u^\top \nabla F_{t,i}(w) \nabla F_{t,j}(w)^\top u \right).$$

Similar to the first part, we introduce a new function that relabels the quantities in the expression. Let $h_2 : \mathbb{R}^{2K^2} \to \mathbb{R}$ be defined as $h_2(a,b) = \sum_{i,j=1}^K a_{i,j} b_{i,j}$. Let $f_1 : \mathcal{W} \times \mathbb{R}^d \to \mathbb{R}^{K^2}$ be given by $f_1(w,u) = (\nabla^2 G_t)(F_t(w))$ and $f_2 : \mathcal{W} \times \mathbb{R}^d \to \mathbb{R}^{K^2}$ be given by $f_2(w,u) = (u^\top \nabla F_{t,i}(w) \nabla F_{t,j}(w)^\top u)_{i,j \in [K]}$. We apply block-wise contraction (Lemma 3)

with one block for $f_1$ and one block for $f_2$ to conclude

$$\frac{1}{2}\mathbb{E}_\epsilon \sup_{w\in\mathcal{W}} \sup_{\substack{u\in\mathbb{R}^d \\ \|u\|_2\leq 1}} \sum_{t=1}^n \epsilon_t h_2(f_1(w,u), f_2(w,u))$$

$$\leq L_{F,2}\,\mathbb{E}_\epsilon \sup_{w\in\mathcal{W}} \sup_{\substack{u\in\mathbb{R}^d \\ \|u\|_2\leq 1}} \sum_{t=1}^n \langle \boldsymbol{\epsilon}_t, f_1(w,u)\rangle + L_{G,2}\,\mathbb{E}_\epsilon \sup_{w\in\mathcal{W}} \sup_{\substack{u\in\mathbb{R}^d \\ \|u\|_2\leq 1}} \sum_{t=1}^n \langle \boldsymbol{\epsilon}_t, f_2(w,u)\rangle$$

$$\leq L_{F,2}\,\mathbb{E}_\epsilon \sup_{w\in\mathcal{W}} \sum_{t=1}^n \langle \boldsymbol{\epsilon}_t, \nabla^2 G_t(F_t(w))\rangle + L_{G,2}\,\mathbb{E}_\epsilon \sup_{w\in\mathcal{W}} \sup_{\substack{u\in\mathbb{R}^d \\ \|u\|_2\leq 1}} \sum_{t=1}^n \sum_{i,j=1}^K \epsilon_{t,i,j} u^\top \nabla F_{t,i}(w)\nabla F_{t,j}(w)^\top u$$

$$= L_{F,2}\,\mathbb{E}_\epsilon \sup_{w\in\mathcal{W}} \sum_{t=1}^n \langle \boldsymbol{\epsilon}_t, \nabla^2 G_t(F_t(w))\rangle + L_{G,2}\,\mathbb{E}_\epsilon \sup_{w\in\mathcal{W}} \left\|\sum_{t=1}^n \sum_{i,j=1}^K \epsilon_{t,i,j} \nabla F_{t,i}(w)\nabla F_{t,j}(w)^\top\right\|_\sigma.$$

Combining the two terms gives the desired chain rule. $\qquad\square$

**Proof of Lemma 12.** As in Lemma 7, let $G_t(s) = (\sigma(s)-y_t)^2$ and $F_t(w) = \langle w, x_t\rangle$, so that $\ell(w\,;x_t,y_t) = G_t(F_t(w))$.

Observe that $G'_t(s) = 2(\sigma(s)-y_t)\sigma'(s)$, $\nabla F_t(w) = x_t$, $\nabla^2 F_t = \mathbf{0}$, $G''_t(s)(s) = 2(\sigma'(s))^2 + 2y_t\sigma''(s)$, and $G'''(s) = 4\sigma'(s)\sigma''(s) + 2y_t\sigma'''(s)$, which implies that $|G'''_t(s)| \leq 6C_\sigma^2$. Using Theorem 10 with constants $L_{F,1} = R^2$, $L_{F,2} = 0$, $L_{G,1} = 2C_\sigma^2$ and $L_{G,2} = 4C_\sigma^2$, we get

$$\mathbb{E}_\epsilon \sup_{w\in\mathcal{W}} \left\|\sum_{t=1}^n \epsilon_t \nabla^2 \ell(w\,;x_t,y_t)\right\|_\sigma \leq 2R^2\,\mathbb{E}_\epsilon \sup_{w\in\mathcal{W}} \sum_{t=1}^n \epsilon_t G''_t(\langle w, x_t\rangle) + 8C_\sigma^2\,\mathbb{E}_\epsilon \sup_{w\in\mathcal{W}} \left\|\sum_{t=1}^n \epsilon_t x_t x_t^\top\right\|_\sigma,$$

applying Lemma 3,

$$\leq 12R^2 C_\sigma^2\,\mathbb{E}_\epsilon \sup_{w\in\mathcal{W}} \sum_{t=1}^n \epsilon_t\langle w, x_t\rangle + 8C_\sigma^2\,\mathbb{E}_\epsilon \sup_{w\in\mathcal{W}} \left\|\sum_{t=1}^n \epsilon_t x_t x_t^\top\right\|_\sigma$$

$$= 12R^2 C_\sigma^2 B\,\mathbb{E}_\epsilon \left\|\sum_{t=1}^n \epsilon_t x_t\right\| + 8C_\sigma^2\,\mathbb{E}_\epsilon \left\|\sum_{t=1}^n \epsilon_t x_t x_t^\top\right\|_\sigma.$$

Invoking Theorem 7 and Fact 1, we have $\mathbb{E}_\epsilon \|\sum_{t=1}^n \epsilon_t x_t\| \leq \sqrt{2\beta R^2 n}$ and $\mathbb{E}_\epsilon \|\sum_{t=1}^n \epsilon_t x_t x_t^\top\|_\sigma \leq \sqrt{2\log(d)R^4 n}$. $\qquad\square$