[Reviews · NeurIPS 2018]

Reviewer 1



The paper provides upper bounds on sample complexity at which the gradients of the empirical loss uniformly converge to the gradients of the population loss. With uniform convergence of empirical gradients to population gradients, one can now argue that for non-convex optimization problems where the population objective satisfies KL (gradient dominance) condition, any algorithm that reaches first order stationary points of the empirical objective indeed returns a solution that has near optimum population loss. The results are applied to get sample complexity results for minimizing population loss in the special cases of generalized linear models and robust regression in high dimensional regime and for one hidden unit ReLU network (under additional margin conditions). The main technical novelty here is that the authors directly manipulate vector valued Rademacher complexities to provide dimension independent bounds on sample complexity. The tools and the results are potentially applicable to wider class of problems. Comments: - The complexity of the model class \mathcal{W} here is controlled by the norm bounds of the predictor B (and associated upper bound R on dual norm of inputs x). In this sense, G and R are key quants of interest in reasoning about sample complexity and explicit dependence on them should be included in the main equations in the theorem and in discussions. - (comment on future work) In order to bound population loss of solutions returned by first order critical points of empirical loss, we only need uniform convergence of gradients at critical points of the empirical loss and not in the entire parameter space. In particular, for regions of the parameter space where the gradients of the empirical loss are large, we do not necessarily need convergence to population gradients. I wonder if this can be exploited to get stronger upper bounds by allowing the upper bound to explicitly depend on ||\nabla L(w)||. ----- Read author response

Reviewer 2



This paper studies the uniform convergence of gradients for non-convex ERM problems. The results are applied to function satisfying Kurdyka-Łojasiewicz conditions and in particular certain classes of generalized linear models, to get risk bounds based on the empirical gradient at the output points. The main tools for the analysis are normed and vector-valued Rademacher complexity. The key technical contribution is a chain rule, which bounds the normed Rademacher complexity of the gradient of composite function, using their Lipschitz parameters and the vector-valued/normed Rademacher complexities of the gradient of each functions. This serves the role of contraction lemma in the proof of uniform convergence of function values. I think this is an okay paper with legitimate contribution, but perhaps not novel enough for NIPS. There are several minor flaws, which further weaken its publishability. The use of contraction-type arguments to bound vector norm Rademacher complexities is interesting. However, the comparison to [33] (Mei, Bai and Montanari, 2016) is not well-justified. In particular, for low-dimensional setup, this paper assumes data points to be uniformly bounded in an $\ell_2$ ball, and the bound has bad dependence on its radius; in [33], assumption on data points is sub-Gaussian on each direction, which gives a $\sqrt{d}$ radius with high probability. If the authors are going to make comparisons, they should rescale their radius parameter to $\sqrt{d}$. Technical comments: 1. The concentration results looked weird. In Proposition 1, they get $O(\log 1/\delta/n)^\alpha$ for $alpha\in[1,2]$, and in Proposition 2, the dependence on failure probability gives $O(\log 1/\delta/n)$. This seems to be much better than standard Hoeffding or McDiarmid inequalities, but in the proof, the author seems to be using nothing more than those. So I don't think these bounds are possible in terms of concentration term. 2. If K-Ł is satisfied, is uniform convergence actually necessary? Does localized empirical process give better rates? Intuitively we can show proximity to population optimal, and only need to do localized arguments within a small ball, which could possibly lead to localized rates.

Reviewer 3



In this paper, the authors studied uniform convergence of gradients based on vector-valued Rademacher complexities and KL conditions of popular risks. With applications to non-convex generalized linear models and non-convex robust linear regression, the authors showed the effectiveness of this uniform convergence to study generalization bounds of any model satisfying approximate stationary conditions. (1) It seems that the gradient uniform convergence in Section 2 follows from standard applications of existing results in the literature. Since Section 2 provides the foundation of the subsequent analysis, I am not sure whether the contribution is enough. (2) For generalized linear models, the authors only considered the specific least squares loss. Is it possible to extend the discussions to more general loss functions? (3) Assumption 1 seems to be a bit restrict. It would be helpful to give some examples of $\sigma$ satisfying (a) and (b). Furthermore, the part (c) requires the conditional expectation to be realized by some $w^*\in\mathcal{W}$, which is a bounded set encoding linear functions. In this case, the learning problems in Section 3 must be relatively easy. This comment also applies to Assumption 2. (4) In Theorem 8, the supremum is taken over a random set $\mathcal{W}(\phi,\hat{\mathcal{D}})$, which seems not natural. In my opinion, it would be interesting to consider the supremum over a non-random set. Minor comments: (1) $\beta$-smooth norm is used in Theorem 3 but the definition is given in the appendix. (2) for any $\delta>0$ should be for any $0<\delta<1$ (3) There are repeated references in [20,21]